# WEBOPERATOR : ACTION-AWARE TREE SEARCH FOR AUTONOMOUS AGENTS IN WEB ENVIRONMENT

## ABSTRACT

LLM-based agents often operate in a greedy, step-by-step manner, selecting actions solely based on the current observation without considering long-term consequences or alternative paths. This lack of foresight is particularly problematic in web environments, which are only partially observable—limited to browser-visible content such as the current page's DOM and UI elements—where a single misstep often requires complex and brittle navigation to undo. Without an explicit backtracking mechanism, agents struggle to correct errors or systematically explore alternative paths. Tree-search methods provide a principled framework for such structured exploration, but existing approaches lack mechanisms for safe backtracking, making them prone to unintended side effects. They also assume that all actions are reversible, ignoring the presence of irreversible actions—limitations that reduce their effectiveness in realistic web tasks. To address these challenges, we introduce **WebOperator**, a tree-search framework that enables reliable backtracking and strategic exploration. Our method incorporates a best-first search strategy that ranks actions by both reward estimates and safety considerations, along with a robust backtracking mechanism that verifies the feasibility of previously visited paths before replaying them, preventing unintended side effects. To further guide exploration, WebOperator generates action candidates from multiple, varied reasoning contexts to ensure diverse and robust exploration, and subsequently curates a high-quality action set by filtering out invalid actions pre-execution and merging semantically equivalent ones. Experimental results on WebArena and WebVoyager demonstrate the effectiveness of WebOperator. Notably, on WebArena, WebOperator achieves state-of-the-art performance with gpt-4o, achieving a 54.56% success rate, underscoring the critical advantage of integrating strategic foresight with safe execution.

## 1 INTRODUCTION

LLM-based WebAgents are increasingly applied to automate complex web interactions, ranging from form filling and content retrieval to multi-step workflows over dynamic pages (Deng et al., 2024). However, planning and executing such tasks remains challenging due to unique characteristics of web environments, such as being *partially observable*: the agent can access the current page's DOM, UI elements, and visible content, but has no direct access to hidden server-side state or the broader global context.

Despite these challenges, conventional WebAgents operate in a greedy, step-by-step manner, selecting actions based solely on the current observation, without accounting for long-term consequences or alternative strategies (Ning et al., 2025). While off-the-shelf models for estimating action usefulness are available, they are inherently short-sighted and imperfect (Chae et al., 2025) and this myopic approach is particularly fragile in web-like, partially observable environments, where a single incorrect action can lead the agent into a state, from which the goal is unreachable. Without an explicit backtracking mechanism, agents struggle to correct errors or systematically explore alternative paths. While tree search provides a principled foundation for decision-making with backtracking, existing methods struggle to generalize across diverse problems and real-world complexities, leaving much of its potential unrealized. In this paper, we introduce **WebOperator**, a performant, action-aware tree-search framework for autonomous agents that systematically addresses these fundamental challenges and operates robustly in dynamic, real-world web settings.

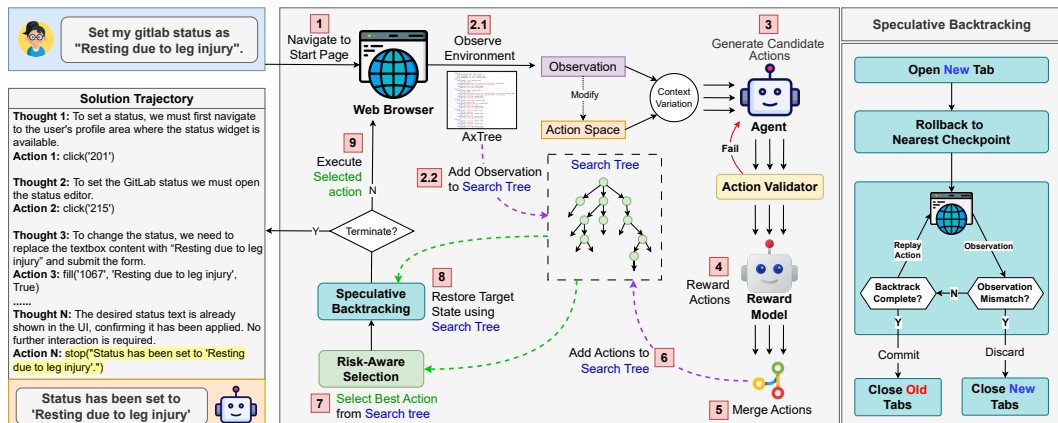

Figure 1: Overview of **WebOperator**, a tree-search framework for solving web tasks. The workflow iteratively explores the web environment via a structured tree: it (1) initializes at the start page, (2) observes and encodes the current page state as a node in the search tree, (3) adapts action space using the current observation, and expands the node by generating candidate actions using varied contextual formulations, and these actions are validated through rule-based analysis and simple URL-existence checks; (4) evaluates actions with a reward model, (5) merges duplicate or equivalent actions, (6) updates the search tree, (7) selects the best unexecuted action using action-aware criteria, (8) restores the target state using speculative backtracking, (9) executes the selected action, and (10) repeats until a terminating action produces a complete solution trajectory. The *left panel* shows an example thought–action sequence produced during task execution, and the *right panel* details the speculative backtracking mechanism. A detailed step-by-step example of the tree search is provided in Appendix (Figure 10).

Designing tree-search WebAgents presents several core challenges that are specific to search-based decision making in realistic web environments: (i) *low-quality action generation* - LLMs may produce invalid or contextually irrelevant actions (e.g., `go_back` on the start page), reducing tree search efficiency and wasting computation. (ii) *redundant candidate actions* - Fixed number of candidate actions are generated at each step, many of which are redundant or semantically equivalent. Multiple actions may lead to the same outcome without contributing new information, reducing meaningful exploration. (iii) *fragile state reversibility* - Tree search fundamentally depends on reversing or replaying actions to explore alternative paths. However, real web environments are non-deterministic: asynchronous updates, DOM mutations, and navigation effects can make naïve backtracking unreliable. Replaying previously executed actions may fail or lead to an inconsistent state, limiting the practical effectiveness of tree search. (iv) *handling destructive actions* - Many real web interactions, such as submitting irreversible forms, deleting items, or logging out, permanently alter the environment. Existing tree search methods (Koh et al., 2024; Zhou et al., 2024; Zhang et al., 2025a) assume reversibility and cannot safely reason about or schedule these destructive actions. Executing them carelessly can invalidate previously visited states, preventing reliable backtracking and limiting exploration of alternative paths. (v) *computational overhead* — exhaustive tree search is prohibitively expensive in realistic web settings. Monte Carlo Tree Search (MCTS), for instance, relies on extensive random rollouts and costly environment resets, making it ill-suited for web-scale (Zhou et al., 2024; Zhang et al., 2025a).

To address these challenges, WebOperator pioneers a redefinition of web environments by extending the notions of state (temporary and persistent) and actions (safe and destructive). It develops an action-aware tree-search approach that incorporates: (a) Dynamic adaptation of the action space at each step based on the current observation, along with validation of generated actions to reject those that are invalid or have no meaningful effect. (b) Variation of the LLM input context to generate diverse candidate actions, combined with consolidation of redundant actions to ensure meaningful exploration. (c) Reliable backtracking using speculative execution and snapshot validation, allowing previously executed actions to be replayed or aborted without corrupting the main environment. (d) Pre- and post-execution heuristics to identify potentially destructive actions based solely on observable content. (e) Efficient traversal via a best-first search strategy that prioritizes safe, reversible actions early and defers destructive actions, replacing costly random-rollout methods like MCTS.

Table 1: Comparison of tree-search WebAgents across key capabilities, highlighting WebOperator.

| Method | Dynamic Action Space | Action Validation | Context Variation | Action Merging | Non-deterministic Environment | Destructive Action Handling |
|---|---|---|---|---|---|---|
| LM-TS (Koh et al., 2024) | ✗ | ✗ | ✗ | ✗ | ✗ | ✗ |
| LA-TS (Zhou et al., 2024) | ✗ | ✗ | ✗ | ✗ | ✗ | ✗ |
| WebPilot (Zhang et al., 2025a) | ✗ | ✗ | ✗ | ✗ | ✗ | ✗ |
| Branch-n-Browse (He et al., 2025) | ✗ | ✗ | ✗ | ✗ | ✗ | ✗ |
| WebRollback (Zhang et al., 2025b) | ✗ | ✗ | ✗ | ✗ | ✗ | ✗ |
| *WebOperator (Ours)* | ✓ | ✓ | ✓ | ✓ | ✓ | ✓ |

Together, these contributions enable WebOperator to systematically explore web environments, safely handle both temporary and persistent state changes, and operate efficiently under uncertainty, advancing the capabilities of tree search for realistic web automation tasks. Table 1 presents a feature-based comparison with prior methods, while Figure 1 provides an operational overview. Through comprehensive experiments on two dynamic, real-world web benchmarks, WebArena and WebVoyager, we demonstrate the effectiveness of WebOperator. Our ablation studies and analyses further provide deeper insights into WebOperator's capabilities and limitations.

## 2 PRELIMINARIES

### 2.1 PROBLEM DEFINITION: TREE SEARCH IN WEB ENVIRONMENTS

Let a **web environment** be represented as a tuple $E = (S, A, T, O)$, where:

- $S$ is the **state space**, partitioned into **persistent state** (e.g., server-side data, cookies, local storage) and **temporary state** (e.g., DOM elements, scroll offsets, open tabs).
- $A$ is the set of **actions** the agent can perform (e.g., click, fill form, navigate, terminate). The complete action set is provided in Table 4 (Appendix).
- $T : S \times A \to S$ is the **transition function**, describing how actions change the environment state. Transitions may be **stochastic** due to dynamic page content.
- $O : S \to \mathcal{O}$ is the **observation function**, mapping states to agent-observable snapshots (DOM, page content, URL, etc.).

A **tree search for web automation** constructs a *search tree* $\mathcal{T}$, where each **node** represents a reachable state $s \in S$ and each **edge** corresponds to an action $a \in A$ that transitions from the parent state to the child state. The agent's goal is to find a **sequence of actions** from the root node (initial state) to a **target node** (goal state).

### 2.2 ACTION CLASSIFICATION

WebOperator classifieds actions based on their impact on the web environment and their reversibility. This classification allows the agent to reason about risks and choose actions that maintain safety and efficiency during tree search.

**Safe Actions.** Actions that only modify *temporary state*, such as scrolling, interacting with drop-downs, or toggling checkboxes. These actions are fully reversible, meaning the agent can return to the previous state without affecting persistent data. Navigational actions, which change the page URL and effectively reset the temporary state while preserving persistent state, are treated as special safe actions. Safe actions (including navigational actions) form the majority of exploratory steps in the tree search.

**Destructive Actions.** Actions that modify *persistent state*, including server-side changes, form submissions, or updates to browser storage and cookies. Destructive actions are high-risk and cannot always be undone. WebOperator carefully considers these actions and applies them only when necessary, with mechanisms to verify safety and prevent unintended consequences.

**Terminating Actions.** Actions proposed by the agent to *terminate the search* at the current node. These actions do not modify the environment but signal that the agent considers the current node as a sufficient solution or stopping point for exploration.

**Invalid Actions.** Actions that are syntactically or semantically incorrect and would *raise an execu-*

*tion error* if performed in the environment. Examples include navigating to an invalid URL, clicking a disabled or non-existent element, filling a read-only field, or attempting to go back from the initial page.

# 3 METHODOLOGY

In this section, we describe our WebOperator framework for robust and efficient tree search in web environments. Building on the task formalization and action taxonomy introduced in §2.1 and §2.2, WebOperator develops a best-first search strategy that integrates several key components: a rich action generation process (§3.1) that combines adaptive action space, context variation, action validation, and action merging to produce high-quality and diverse candidates; an improved backtracking procedure (§3.3) to ensure efficient and reliable traversal; specialized handling of destructive actions (§3.2), which are the primary source of environment corruption; and a dynamic, context-aware action selection mechanism (§3.4) that balances reward, safety, reversibility, and search context while maintaining a bounded frontier. The complete search algorithm, integrating all these components, is provided in Appendix D..

## 3.1 ACTIONS GENERATION

Candidate actions are generated using a large language model (LLM) based on a rich contextual representation of the current state. WebOperator addresses the two primary challenges: low-quality and redundant actions.

To generate **high-quality actions**, WebOperator combines proactive and reactive measures:

- **Dynamic Action Space.** The set of available action types is dynamically adapted to the current observation, ensuring only feasible actions are considered at each step (e.g., `go_back` is allowed only when there are previous pages). This reduces irrelevant exploration and prevents invalid actions.
- **Action Validation via Error Prediction.** Each generated action is checked before execution. Static analysis inspects the DOM/accessibility tree (e.g., element visibility, enabled status), while simple dynamic checks ensure that actions such as navigation target valid URLs. Actions that are invalid or ineffective are rejected, and feedback is provided to the LLM to regenerate improved candidates.

To generate **diverse actions**, WebOperator further employs complementary proactive and reactive strategies:

- **Context Variation.** Different components of the LLM input are varied for each candidate action, such as the length of task history included or selective retrieval of relevant past trajectories. This encourages the model to propose semantically distinct actions.
- **Action Merging.** Semantically equivalent actions are consolidated after generation to avoid redundant expansions, effectively reducing the branching factor and ensuring meaningful exploration.

This combination of proactive and reactive mechanisms ensures that the candidate actions entering the search are both high-quality and diverse, directly addressing the core limitations of traditional tree-search WebAgents. Detailed implementation of each component is provided in Appendix A.1.

## 3.2 THE CHALLENGE OF DESTRUCTIVE ACTIONS

A key challenge in web tree search is handling actions that permanently modify the environment, i.e., **destructive actions**. Executing such actions may alter the database or browser state (cookies, local storage, session storage), potentially invalidating[1] previously visited states and corrupting the main environment.

**Pre-execution Heuristic for detecting Destructive Actions.** Before executing an action, WebOperator employs a lightweight heuristic to proactively identify potentially destructive operations. This

---

[1]Invalid state means we can't backtrack to that state with our proposed backtracking algorithm

heuristic considers both the action type and the interacted element. Non-click actions (e.g., scroll, tab focus) are generally non-destructive. For clicks, only `button` elements are considered potentially destructive, while links and other elements are typically safe. Similarly, pressing the `Enter key`, which often triggers form submissions or default button actions, is treated as potentially destructive. Buttons with labels indicating navigation or transient actions (e.g., back", search", "refresh") are classified as non-destructive. The full detection procedure is summarized in Algorithm 2 (Appendix).

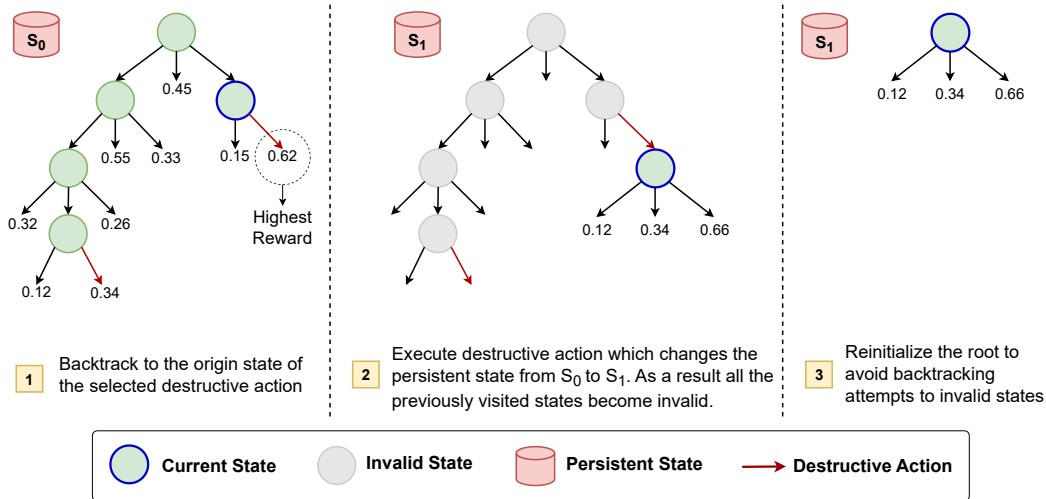

Figure 2: Illustration of destructive action execution in WebOperator

**Post-execution Heuristic for detecting Destructive Actions.** While the pre-execution heuristic provides an initial estimate based on static cues, it lacks runtime information available only after execution. To address this limitation, we introduce a post-execution heuristic that leverages network-level observations to validate whether an executed action was indeed destructive.

During execution, WebOperator monitors the network activity triggered by the action, analyzing the corresponding HTTP requests. Actions resulting in `GET` requests are typically non-destructive, as they merely retrieve data. In contrast, actions generating `POST`, `PUT`, `DELETE`, or `PATCH` requests are strong indicators of destructive operations, as they usually modify server-side data or trigger irreversible effects.

After execution, WebOperator applies this heuristic to reassess the action's destructiveness. If confirmed, all previous states are invalidated, and the search tree is reset as described below. This two-stage detection mechanism—combining pre- and post-execution heuristics—provides both proactive avoidance and reactive correction, ensuring robust handling of destructive behaviors throughout the search process.

**Handling Destructive Actions' Execution.** When an action is detected as destructive, its execution marks a point of no return in the search process. Once the environment's persistent state changes, previously visited states may become inconsistent. Continuing to expand or backtrack to these states could lead to invalid or unsafe transitions.

WebOperator handles this situation as follows:

1. **Invalidate All Previous States:** All states in the tree, except the current states resulting from the destructive action, are marked as invalid and excluded from further expansion.

2. **Reset Tree Root:** The current state becomes the new root of the search tree.

3. **Resume Exploration:** Tree search continues from this new root using the action-aware Best-First Search, generating new candidate actions based on the updated environment state.

This mechanism ensures that exploration can safely continue even after executing destructive actions, preserving the correctness of the search and preventing invalid backtracking.

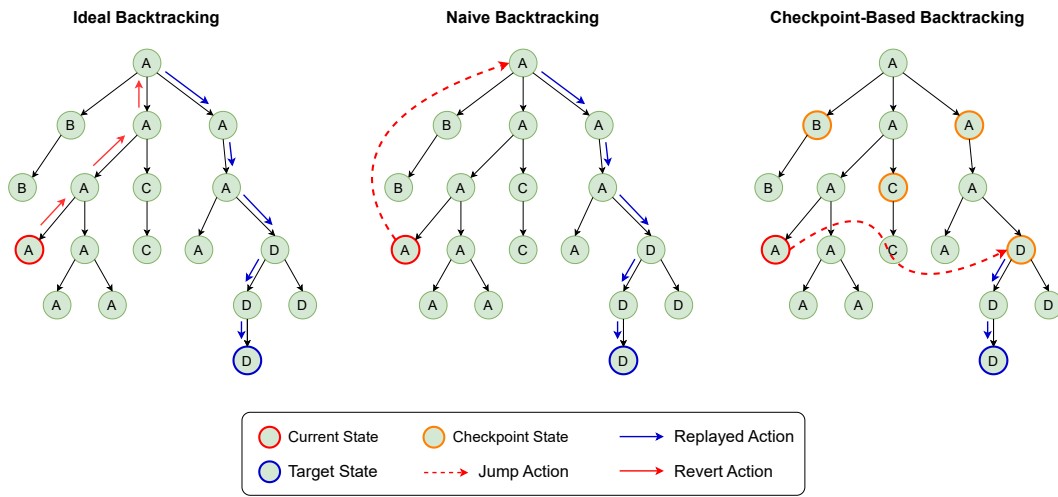

Figure 3: Different types of Backtracking visualization. See Appendix E for a detailed comparison.

## 3.3 BACKTRACKING

Backtracking is a crucial operation in tree search for web automation, where the agent may need to execute an action originally generated from a previous state rather than the current one. Formally, backtracking navigates the agent from the current state to a prior state where a promising action was proposed, typically by (1) restoring the environment to an ancestor of the target state, and (2) executing the minimal sequence of actions required to reach the target. Prior works (Koh et al., 2024; He et al., 2025) rely on a naive approach that resets the root state and replays all actions to reach the target, which is highly inefficient. More critically, as discussed earlier, executing destructive actions can invalidate previously saved states, including the original root and many intermediate ancestors — in such cases resetting to that root for backtracking is impossible or unsafe. To address both issues, WebOperator introduces an improved backtracking mechanism that addresses both efficiency and reliability concerns.

**Efficiency.** To reduce replay overhead, WebOperator employs *checkpoint-based state jumping* (illustrated in Fig. 3). A state is marked as a checkpoint when its webpage observation remains unchanged under refresh and its URL differs from that of its parent. Such states are safe to revisit directly because they are refresh-stable and represent distinct navigation points, ensuring that jumping to their URL reliably reconstructs the same underlying environment without requiring prior UI interactions. During backtracking, the agent jumps to the closest checkpoint of $s_t$ via URL navigation, then replays only the minimal UI interactions (e.g., scrolling, form filling) needed to reconstruct $s_t$. This strategy avoids unnecessary intermediate steps by leveraging the determinism of URL-based navigation: navigate to the nearest checkpoint, then apply only the incremental UI operations required to recover the exact target state.

**Reliability.** To handle non-deterministic behaviors, WebOperator employs *speculative backtracking* with *snapshot validation*. Instead of replaying actions directly in the main environment, the agent attempts backtracking in a parallel browser tab. Before executing each stored action, the agent compares the current observation in the parallel tab with the corresponding stored snapshot[2] of the target state $s_t$ (see Appendix F for the comparison procedure). If any mismatch indicates that the state is no longer reproducible—due to randomness, dynamic content changes, or UI drift—the backtracking attempt is immediately aborted, leaving the main environment unchanged. If all actions replay successfully and all snapshots match, the reconstructed state is committed to the main environment. This speculative execution prevents unintended side effects and ensures reliable state restoration even in non-deterministic web environments.

---

[2]Observation stored in the search tree during the first visit

Table 2: Success rate (SR %) comparison of Tree Search Agents on WebArena with gpt-4o *(BFS = Best-First Search)*.

| Agent | Search Algorithm | Search Budget | Overall (#812) | Reddit (#106) | GitLab (#180) | Shopping (#187) | CMS (#182) | Map (#109) | Multisite (#48) |
|---|---|---|---|---|---|---|---|---|---|
| LM-TS | BFS | 20 / Iteration | 19.2 | 11.3 | 13.9 | 27.8 | 16.5 | 26.6 | 16.7 |
| Branch-n-Browse | BFS | 10 / Sub-task | 35.8 | 50.9 | 36.7 | 34.6 | 26.4 | 46.8 | 18.8 |
| WebPilot | MCTS | 10 / Sub-task | 37.2 | 65.1 | 39.4 | 36.9 | 24.7 | 33.9 | - |
| *WebOperator* | BFS | 5 (Overall) | 24.4 | 15.1 | 20.0 | 31.0 | 21.4 | 39.5 | 12.5 |
| | | 10 (Overall) | 42.7 | 63.2 | 37.2 | 42.8 | 40.1 | 46.8 | 18.8 |
| | | 15 (Overall) | 48.4 | 71.7 | 42.2 | 46.0 | 47.3 | 52.3 | 25.0 |
| | | **20 (Overall)** | **54.6** | **76.4** | **52.8** | **49.2** | **55.0** | **55.2** | **31.3** |

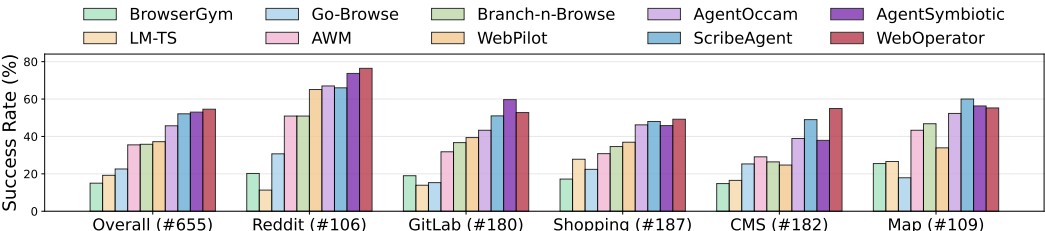

Figure 4: Comparison of success rates between WebOperator and the baseline across different task categories (see Table 5 in Appendix).

## 3.4 ACTION SELECTION

After each expansion step, newly generated candidate actions are scored and inserted into a *frontier*—a priority queue of unexecuted actions that the tree search may choose from. Unlike prior approaches, which select the next action solely based on these static scores, WebOperator employs a *dynamic, context-aware selection policy*. At every step, the priority of each frontier action is recomputed based on (i) its type (safe, destructive, terminating, or repetitive) and (ii) the evolving search context, such as progress toward the goal, past destructive actions, or accumulated step count. This allows the agent to adaptively steer exploration: safe and reversible actions are favored early, destructive actions are deferred until they are strategically justified, and terminating actions are promoted only when they represent high-confidence opportunities.

To keep the search tractable, WebOperator maintains a fixed *frontier budget*. Whenever the frontier exceeds this budget, it is pruned through a structured reduction process. First, actions that cannot be reliably backtracked to are removed. Next, among destructive actions, only the highest-rewarded one is retained, since once a destructive action is selected, all others become irrelevant—the search tree is reset afterward. A similar rule is applied to terminating and repetitive actions. If the frontier still exceeds its budget, semantically duplicate safe actions are merged, and any remaining overflow is resolved by discarding the lowest-rewarded safe actions.

Through this dynamic prioritization and controlled frontier management, WebOperator selects actions not merely by reward, but by their safety, reversibility, contextual relevance, and contribution to meaningful long-horizon exploration. The complete procedure is detailed in Algorithm 5.

## 4 EXPERIMENTS

Here, we detail experiments on WebArena (Zhou et al.), a web simulator benchmark. Further experiments with WebVoyager (He et al., 2024), a web benchmark based on real-world websites, are included in Appendix C.2.

## 4.1 EXPERIMENTAL SETUP

**Environment.** We utilize WebArena, an interactive web simulator benchmark comprising fully functional websites across four domains: e-commerce (OneStopShop), social forums (Reddit), col-

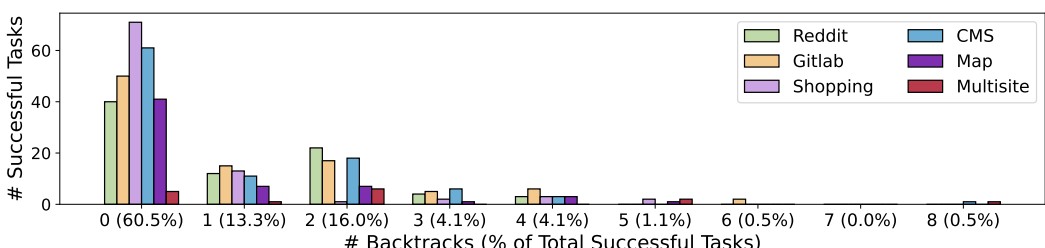

Figure 5: Distribution of successful WebArena tasks grouped by the number of backtracks required across different task domains.

laborative software development (GitLab), and content management (online store management). The environment also provides utility tools (map, calculator, scratchpad, Wikipedia) to support realistic interactions. WebArena contains 812 tasks instantiated from 241 parameterized templates, each paired with a programmatic evaluator that verifies task completion against ground-truth targets.

**Implementation.** We implement WebOperator on top of the BrowserGym framework (Drouin et al.). The backend language model is `gpt-4o-2025-01-01`, serving as the core reasoning and evaluation module throughout action generation, and scoring. Unless otherwise stated, the tree search uses a depth factor $d = 5$, frontier budget 4, and branching factor $b = 3$ with a search budget of 20 steps per task.

**Baselines.** We compare WebOperator with the following prior and concurrent tree search methods: (1) LM-TS (Koh et al., 2024), (2) WebPilot (Zhang et al., 2025a), (3) Branch-and-Browse (He et al., 2025). All baselines use gpt-4o as their backbone, and we therefore directly report the results from their respective papers.

### 4.2 Main Results

Table 2 presents our main results on WebArena, comparing WebOperator against tree search baselines. With a search budget of 20, our method achieves a 54.6% overall success rate, substantially outperforming Branch-n-Browse (35.8%) and WebPilot (37.2%). Crucially, even with a lower budget of 10, WebOperator (42.7%) already outperforms all existing tree-search methods that use higher per-task budgets. This demonstrates superior budget efficiency, with performance scaling consistently from 24.4% (budget 5) to 54.6% (budget 20). The method also shows strong cross-domain performance, excelling on Reddit (76.4%), GitLab (52.8%), and CMS (55.0%) tasks. A complete comparison, including non-tree-search methods, is shown in Figure 4.

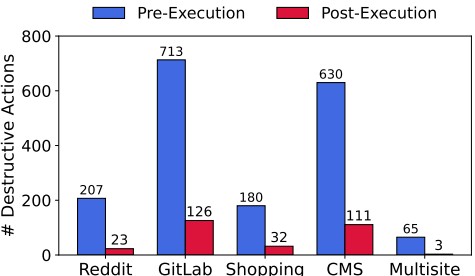

Figure 6: Comparison of number of destructive actions detected by pre- and post-execution heuristic across different task categories (Map is excluded since destructive action is not applicable for read-only websites).

Figure 6 further shows that WebOperator's pre- and post-execution heuristics are key to achieving these gains in a safe yet exploratory manner. The pre-execution heuristic conservatively flags potentially destructive actions, while the post-execution check later confirms which of them actually modify persistent state, with only about 16% of pre-flagged actions turning out to be truly destructive. Together, these mechanisms allow the agent to remain exploratory without being overly cautious, and to reliably reset and re-root the search tree when irreversible changes occur.

As shown in Figure 5, while a majority of tasks are solved without any backtracking, approximately 40% of successful tasks required at least one backtrack, highlighting the importance of WebOperator's robust corrective search mechanism. Tasks requiring 5 or more backtracks remain rare ($< 3\%$), indicating that extreme corrective efforts are uncommon in WebArena. These patterns emphasize both the efficiency and reliability of WebOperator: it can handle straightforward tasks quickly while

Table 3: Success rate (SR %) comparison on WebArena-lite using gpt-4o.

| Agent | Avg Generated Actions | Success Rate (SR %) | | | | | |
|---|---|---|---|---|---|---|---|
| | | Overall (#155) | Reddit (#19) | GitLab (#30) | Shopping (#45) | CMS (#35) | Map (#26) |
| Base ReAct Agent | 9.30 | 47.74 | 57.89 | 56.67 | 46.67 | 31.43 | 53.85 |
| + Dynamic Action Space | 9.17 | 49.03 | 52.63 | 63.33 | 46.67 | 40.00 | 46.15 |
| + Action Validation | **8.67** | 53.55 | 68.42 | **70.00** | 51.11 | 42.86 | 42.31 |
| + Multi-Action | 24.06 | 52.90 | 68.42 | 63.33 | 48.89 | 34.29 | 61.54 |
| + Action Merging | 25.39 | 54.19 | 57.89 | 66.67 | 51.11 | 37.14 | **65.38** |
| + Context Variation | 25.30 | 54.84 | 63.16 | 66.67 | 53.33 | 37.14 | 61.54 |
| + Tree Search | 24.79 | 51.61 | 42.11 | 66.67 | 51.11 | 37.14 | 61.54 |
| + Destruction-Aware | 27.09 | 51.61 | 63.16 | 66.67 | 51.11 | 37.14 | 46.15 |
| + Selection Heuristic | 29.67 | 58.71 | 68.42 | **70.00** | **57.78** | 45.71 | 57.69 |
| + Speculative-Backtracking | 31.34 | **60.00** | **78.95** | **70.00** | 53.33 | **51.43** | 57.69 |

still effectively recovering from mistakes in harder scenarios, contributing to its strong overall and cross-domain performance.

## 4.3 ABLATION STUDIES

We conduct detailed ablation studies on WebArena-lite (Liu et al.), a curated subset of 155 tasks from WebArena, to systematically evaluate the contribution of each component in our WebOperator framework. The progressive integration of components, as shown in Table 2, follows our architectural design.

**Single-Action Agent Enhancement.** We begin with a Base ReAct Agent that generates a single action at each step (47.74% SR). First, we integrate our core interaction improvements: adding a Dynamic Action Space provides a modest gain (49.03% SR), while incorporating Action Validation further improves success rate to 53.55% and significantly enhances efficiency, achieving the lowest average action count (8.67). This validates that filtering invalid actions early is crucial for robust single-step decision-making.

**Transition to Multi-Action Generation.** Building upon the validated single-action components (Dynamic Action Space + Action Validation), we transition from single to multiple action generation. The Multi-Action approach forms the foundation for tree search, though it naturally increases action exploration (24.06 avg. actions). We then enhance this with Action Merging to consolidate redundant actions and Context Variation to diversify exploration, progressively improving the success rate to 54.84%. This demonstrates that diverse, consolidated action proposals facilitate more effective state space exploration, with a slight increase in the average action count (+1.24).

**Advanced Search with Backtracking.** Introducing naive Tree Search to the multi-action framework reduces performance to 51.61% from 54.84%. This demonstrates that basic backtracking alone is insufficient for complex web tasks. The subsequent integration of our advanced reasoning modules proves essential: incorporating Destructive-Action Handling and Context-Aware Action Selection Policy recovers and substantially improves performance to 58.71%. The complete system with Speculative-Execution achieves the highest success rate of 60.00%, confirming that sophisticated action selection and reliable backtracking are essential for maximizing tree search benefits in long-horizon tasks. Notably, while our full approach (31.34 actions) generates +6.55 more actions than naive tree search (24.79 actions), it delivers a substantially higher absolute gain of +8.39% in success rate, confirming that our sophisticated action selection effectively trades moderate computational exploration for significantly improved task completion.

## 5    RELATED WORK

LLM-based web agents increasingly adopt tree-search planning to overcome the brittleness of greedy step-by-step execution. However, existing systems still struggle with key challenges of the web environment—including non-determinism, partial observability, destructive actions, and reliable state recovery.

**Tree-Search Web Agents.** A growing line of work applies tree-search planning to web automation, aiming to overcome the brittleness of greedy LLM execution. LM-Tree Search (LMTS) (Koh et al., 2024), LATS (Zhou et al., 2024), Branch-and-Browse (He et al., 2025), WebPilot (Zhang et al., 2025a), and WebRollback (Zhang et al., 2025b) all adopt the general strategy of expanding multiple candidate actions, evaluating them with learned value functions or heuristics, and traversing the search space through backtracking-based exploration. These systems demonstrate that branching exploration improves robustness on long-horizon tasks, but they still depend heavily on replay-based restoration, limited action validation, and heuristic handling of dynamic or irreversible transitions. Our work builds on this direction by introducing a more principled treatment of destructive actions, higher-quality and more diverse action generation, and a backtracking mechanism designed for partially observable, non-deterministic web environments.

**Safety and Destruction Handling.** A parallel strand of research focuses on safety-aware planning and recovery for LLM-based web agents. WebGuard (Zheng et al., 2025) introduces a classifier trained to predict the risk level of web actions (e.g., SAFE/LOW/HIGH) and uses it to block or flag potentially harmful interactions. InferAct (Fang et al., 2024) simulates the effect of candidate actions using a secondary LLM and warns users before executing risky commands. While effective, these systems generally operate outside the core planning loop—either through external classifiers or simulation-based safety checks—rather than integrating safety guarantees directly into search. In contrast, our method incorporates both pre- and post-execution heuristics for destructive action detection and speculative backtracking, providing recovery guarantees as part of the tree-search framework itself.

## 6    CONCLUSION

In conclusion, WebOperator introduces a risk-aware tree search framework that fundamentally advances autonomous web agents by systematically addressing the challenges of partial observability, non-determinism, and irreversible actions. Through its integrated approach—featuring high-quality action generation, reliable speculative backtracking, and strategic handling of destructive operations—the method enables robust and efficient exploration. The framework achieves state-of-the-art performance on WebArena and demonstrates strong, superior generalization on WebVoyager, establishing itself as a principled solution for reliable web automation that strategically balances exploration with safety and long-term foresight.

## REPRODUCIBILITY STATEMENT

To support reproducibility, we will release the full implementation of WebOperator in the near future. The release will include source code, configuration files, and scripts to reproduce all experiments reported in this paper.

Specifically, we will provide: (i) an implementation built on the open-source BrowserGym framework, (ii) benchmark configurations for WebArena, WebArena-lite, and WebVoyager following official evaluation protocols, (iii) hyperparameter settings including search budget, frontier budget, and thresholds for destructive/terminating actions, and (iv) logs of experiments for verification of reported results.

All experiments can be reproduced using the provided scripts with minimal setup once the repository is released.

LIMITATIONS

Despite the effectiveness of our tree search framework, there are several limitations and considerations for future work:

**Highly Dynamic Environments:** In highly dynamic or non-deterministic websites, speculative backtracking may always fail and eventually making the process a sequential search.

**Destructive Action Heuristics:** While effective in many cases, these heuristics may fail for complex or unconventional interactions, leading to potential irreversible changes in the environment. However, even when the heuristic detection fails, the speculative backtracking mechanism serves as a reliable backup to maintain search safety.

**Reward Model Dependency:** Our approach depends on the process reward model to evaluate candidate actions before execution. The overall performance is influenced by the accuracy and generalization of this model, which may not capture all edge cases.

**Frontier Budget Constraints:** The action queue has a limited size, which constrains exploration. Although we incorporate action merging to improve diversity, very large or complex websites may still require more exploration than the current budget allows.

**Termination Risk:** Selecting terminating actions is inherently risky. Incorrect early execution can end the search prematurely, and while we mitigate this by deferring terminating actions, there is no formal guarantee against mistakes.

These limitations highlight avenues for future work, including more robust destructive action detection, handling highly dynamic pages, improving process reward models, and extending the framework to multi-user or collaborative environments.

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

## A  ACTION GENERATION AND SELECTION

### A.1  INPUT PROCESSING

#### A.1.1  REPHRASED INSTRUCTION

To improve LLM understanding and performance, we reformulate the user instruction into a more LLM-friendly version. This involves clarifying ambiguities, emphasizing important details, and highlighting potential edge cases the model should consider. The rephrased goal helps guide the agent's reasoning and ensures that the input is structured in a way the model can interpret reliably. The complete prompt template used for generating rephrased instructions is shown in Listing 3.

#### A.1.2  ADAPTIVE OBSERVATION SPACE

The agent receives a flattened representation of the accessibility tree as input. However, this alone can make it difficult to determine whether the page is scrollable or whether scrolling would be beneficial. We adapt the observation space based on the size of the accessibility tree and context window constraints:

- **Small accessibility trees:** When the tree is relatively small and fits within the model's context window, the entire page is provided as input, and scrolling is disabled. This ensures the agent has complete context without needing to scroll.
- **Large accessibility trees:** When the tree is large or would exceed the model's context window, only visible elements in the current viewport are provided as input, and scrolling is enabled. This prevents the agent from being overwhelmed while still allowing exploration of off-screen elements.

This strategy balances context completeness with tractability, enabling better decision-making for actions such as scrolling while respecting the model's input limitations.

#### A.1.3  INCORPORATING ALERTS INTO THE ACCESSIBILITY TREE

JavaScript 'alert' dialogs produce pop-ups in the browser that are not part of the standard accessibility tree. However, these alerts can contain important information, such as error messages or confirmation prompts, which may affect task progress or decision-making.

To address this, we detect alerts during execution and include them as nodes in the accessibility tree. This allows the agent to reason about their content and handle them appropriately, integrating these previously hidden signals into action generation.

#### A.1.4  CONCISE HISTORY REPRESENTATION

A major challenge in web navigation is the large context requirement: the agent needs to remember previous steps to make informed decisions, but including the full trajectory (observations, thoughts, actions) quickly exceeds the model's context window.

To address this, we use a concise trajectory representation:

- At each step, the agent is prompted to generate a summary of the current observation.
- In the context for subsequent steps, we provide only the summaries of the last N steps (observation summary, thought, action) rather than the full raw observations.

This approach allows the agent to maintain a meaningful memory of the trajectory while keeping the context size manageable. By summarizing observations, we provide a compact yet informative history, enabling the agent to reason over previous steps without overwhelming the model.

#### A.1.5  INTEGRATING PAST EXPERIENCE

Humans rarely interact with websites purely by exploration; they rely on past experiences, such as remembering how a login form was filled or which sequence of clicks led to a desired page. Inspired by this, we enhance input formation with **retrieval-based context**, allowing the agent to incorporate

relevant prior interactions from a **static database of past trajectories** collected in advance, as in Go-Browse (Gandhi et al., 2025). Purely context-based inputs often fail to capture repetitive interaction patterns, e.g., filling out a profile form or navigating to settings, which typically follow similar sequences even if minor details like button labels or DOM structures change. At each step, the agent encodes the current observation, retrieves the $k$ most similar (observation, thought, action) tuples from the database, and includes them in the prompt. This enriches the model's input with relevant past experience, improving efficiency, robustness to minor UI variations, and producing more human-like reasoning by leveraging prior knowledge instead of relying solely on immediate context.

### A.1.6 DYNAMIC ACTION SPACE

In web environments, not all actions are valid in every state. To improve agent efficiency and reduce irrelevant exploration, we introduce a conditional action space, where certain actions are only available under specific conditions:

- **scroll** – Enabled only when the observation contains visible elements that can be scrolled.
- **select_option** – Enabled only when there is at least one option element present in the observation.
- **tab_close, tab_focus** – Enabled only when multiple tabs are open.
- **go_back, go_forward** – Enabled only when navigation history allows moving backward or forward.

By conditioning the action space in this way, the agent is prevented from generating actions that are impossible or meaningless in the current state. This not only reduces the complexity of decision-making but also improves context relevance and reduces uninformative errors in the trajectory.

Table 4: The adaptive action space of WebOperator .

| Category | Action Type | Description | When Applicable? |
|---|---|---|---|
| **Basic Actions** | click | Click at an element | Always |
| | fill | Fill an element | Always |
| | scroll | Scroll up and down | Page is too long |
| | select_option | Choose an option from a dropdown or select menu | Dropdown/select element is present |
| **Tab Operations** | tab_focus | Focus on the i-th tab | Multiple tabs are open |
| | new_tab | Open a new tab | Always |
| | tab_close | Close current tab | Multiple tabs are open |
| **Page Operations** | go_back | Visit the last URL | Previous page exists in history |
| | go_forward | Undo go_back | Previous action was go_back |
| | goto | Go to URL | Always |
| **Workflow Management** | stop | Stop with an answer | Always |

### A.2 OUTPUT PROCESSING

At each decision step, the agent produces a structured output capturing its reasoning and interaction with the environment. Specifically, the output consists of three components:

- **Thought:** The agent's internal reasoning, capturing why it chooses a particular action.
- **Action:** The concrete operation to perform in the environment, such as clicking a button or entering text.
- **Observation Summary:** A concise representation of the current state, highlighting relevant elements and changes for subsequent reasoning steps.

This structured format ensures that the model's outputs are interpretable and can be fed back into future steps. By explicitly separating reasoning, action, and environment summary, the agent maintains a clear and reusable decision trace throughout the interaction.

### A.2.1 ACTION VALIDATION LOOP

Some generated actions may lead to execution errors, making them uninformative for the trajectory. To address this, we introduce a predictive action validation mechanism that filters likely failures before executing them in the main environment. Validation is performed using a hybrid approach:

- **Simple checks:** For most actions, validation is done using the current accessibility/DOM tree or simple heuristics (e.g., checking if a button is disabled or a field is read-only). These checks are lightweight and do not modify the environment.
- **Speculative tab checks:** For complex cases where success cannot be determined from the DOM alone (e.g., verifying whether a URL is valid), the framework opens a new tab in the same browser context. This preserves sessions and authentication. The URL is loaded, the page is inspected, and then the tab is closed, leaving the main environment unchanged.

Only actions predicted to succeed are executed in the agent's working environment. If an action fails validation, feedback is provided to the agent and an internal retry is allowed. Importantly, these feedback-retry steps are not recorded in the trajectory, keeping the history clean and informative.

Common error scenarios include:

- Generated actions that are syntactically invalid or not included in action space.
- Actions with incorrect parameters.
- Attempts to fill read-only fields or click disabled buttons.
- Attempts to interact with elements that do not exist in the current observation.
- Attempts to access invalid URLs.

### A.2.2 AUTO CORRECTION

While generating actions, agent sometimes make minor mistakes which can easily be corrected without retrying. For example, if agent generates multiple action instead of a single action, WebOperator chooses the first one. Additionally, if agent makes mistake in the syntax of an action, for example generates `fill('24', 'Hello World', false)` instead of `fill('24', 'Hello World', False)`, WebOperator automatically capitalize the `false` to `False`.

### A.2.3 RECOVERY ASSISTANCE

WebOperator tries to avoid errors by Action validation, but despite that even if an execution error occurs it provides detailed feedback to agent. Also, in some cases uses predefined actions. For example, if an invalid page appears (e.g., 404) WebOperator uses the action `go_back` or `tab_close` based on the context. Also, if agent faces *captcha*, WebOperator allows human intervention to escape from that.

### A.2.4 GENERATING MULTIPLE CANDIDATE ACTIONS

At each step, relying on a single action can limit coverage and increase the risk of failure. Generating multiple candidate actions instead allows the agent to compare alternatives and select the most promising path.

Simply sampling multiple times from the same prompt often yields low diversity, since all samples share identical context. To improve variety, we systematically vary the input itself—for example, by adjusting the amount of navigation history included, injecting past experiences via retrieval, or rephrasing the task goal to reduce ambiguity. Such input variations produce richer and more diverse candidate actions, improving robustness across different decision points.

### A.2.5 PROCESS REWARD MODEL FOR CANDIDATE ACTIONS

At each step, multiple candidate actions must be evaluated before deciding which one to execute. To guide this selection, we employ a **process reward model**, which assigns a reward or score to each action based on its likelihood of being productive, *without actually executing it in the environment*.

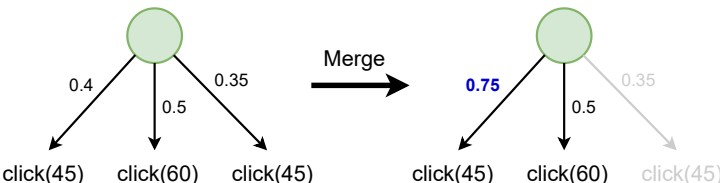

Figure 7: Action merging strategy

Formally, let $a_t$ be a candidate action at time $t$. The process reward model estimates a value $v_t \in [0,1]$ that predicts the expected utility of executing $a_t$. Since the full environment state $s_t$ may not be accessible to the agent (it can include private information such as database entries or browser cookies), the model computes $v_t$ based on observable information:

$$v_t = f_v\Big(I, \{(o_1, a_1), (o_2, a_2), \ldots, (o_t, a_t)\}\Big)$$

where:

- $I$ is the natural language instruction or task goal.
- $o_i$ and $a_i$ are the observation and action at step $i$.
- $f_v$ is the process reward function that predicts the likelihood of an action being productive.

Following **WebShepherd** (Chae et al., 2025), we adopt a checklist-based reward model. From the task instruction and initial observation, a checklist of sub-goals or expected steps is generated. Candidate actions are then evaluated based on how well they contribute to completing items on this checklist. This approach allows the agent to assign rewards to actions without executing them, providing a structured and interpretable measure of expected productivity, which can be used to efficiently rank and select actions.

### A.2.6   ACTION MERGING FOR EFFICIENT EXPLORATION

Multiple candidate actions may target the same element or have the same effect, leading to redundant exploration if treated independently. We implement **action merging** to address this: semantically equivalent actions are identified and their predicted rewards are combined (e.g., summed) before selection. For example, if actions $(A, r_A = 0.4)$, $(B, r_B = 0.5)$, and $(C, r_C = 0.4)$ are generated and $A$ and $C$ are equivalent, merging gives $A + C \to 0.8$ vs $B \to 0.5$, prioritizing the collectively stronger option. This improves efficiency, reduces redundant exploration, and favors actions with higher overall likelihood of success. WebOperator merge actions based on 4 criteria.

1. If two actions are exactly same. Which means function name and all parameters.

2. For multiple `stop` actions, we can safely merge them as only the highest rewarded action is considered for termination.

3. For multiple `fill` actions, if the input text becomes the same after normalizing, we will merge them.

## B   EXPERIMENT DETAILS

### B.1   MODEL

We have used GPT-4o for our experiments. We have used temperature 0.7 for action generation.

### B.2   RETRIEVAL

We retrieving past experience we first need a database where each row contains goal,current observation,though,action.   For that we have used the dataset from Go-Browse (Gandhi

et al., 2025), which proposes a method for automatically collecting diverse and realistic web agent data at scale through structured exploration of web environments. We first needed to preprocess the dataset as the urls in the observation depends on how we host webarena. For example, base url of Reddit website in Go-Browse is `http://ec2-3-148-123-246.us-east-2.compute.amazonaws.com:9999`, which we needed to modify to `http://localhost:9999`, to make it consistent with out setup. To implement semantic retrieval we have used `all-MiniLM-L6-v2` (Wang et al., 2020) as the embedding model. The query contains (goal,observation) and we include the (goal,thought,action) as examples in the context of agent.

### B.3 Action Generation

We have used 3 prompt variations to generate a maximum of 3 actions at each step. After generating an action, that is validated by Action Validator. If an action fails validation, feedback is given to agent and need to generate action again. Each agent need to generate a valid action within 5 retries. After that it will lead to failure. If all agents become successful, we will get 3 valid actions otherwise we will get less. This is a fail-safe architecture, as even if one agent fails other can cover-up that. Also, even if all agent fails at any step we can backtrack and explore other parts of the search tree.

### B.4 Process Reward Model

#### B.4.1 Checklist Generation

Following WebShepherd, WebOperator first generates a checklist that outlines key intermediate milestones for achieving the user's goal. Given an instruction $I$, it produces a checklist $C$ comprising a sequence of natural language subgoals $(g_1, g_2, \cdots, g_k)$. This checklist then serves as the foundation for reward prediction, enabling WebOperator to track progress toward the goal. Similar to action generation we use GPT-4o to generate checklists. A sample checklist for illustration is provided in Listing 5. Also, the prompt for checklist generation is provided in 4.

#### B.4.2 Reward Generation

The generated checklist is then used by the same model to estimate the agent's progress, providing the reward signal $R(s_t)$ that guides the tree search. In our experiments, we have used the same model for both checklist and reward generation, but it can be different. Since the reward is predicted via token generation, the output resides in a discrete space. To obtain a continuous reward signal, several mapping strategies can be employed. Following WebShephered, we employ a verbalizer (Hu et al., 2022) to estimate soft probabilities over label tokens (e.g., "Yes", "No", and "In Progress") using the logits from the LM head. At inference time, WebShephered generate the feedback $F \sim P(\cdot|I, C, o, a)$ and compute the reward for each checklist item using the probabilities of "Yes" and "In Progress" tokens follow:

$$r_k(o, a) = \frac{1}{L} \sum_l^L P(\text{"Yes"}|I, C, o, a, F) + 0.5 \times P(\text{"In Progress"}|I, C, o, a, F), \quad (1)$$

where $L$ denotes the number of checklist and $r_k$ is the score assigned to the $k^{\text{th}}$ response. The final reward is computed as the average: $r(o, a) = \sum_{k=1}^K r_k(o, a)$. Prompt used to assign rewards is shown in 6.

## C Additional Results

### C.1 Additional Baselines on WebArena

In addition to the tree-search baselines reported in the main paper, we include a broader comparison with recent state-of-the-art agents in Table 5. This extended benchmark covers both proprietary and open-source models, as well as non–tree-search approaches. The results reveal that WebOperator maintains state-of-the-art performance across most website categories, particularly on high-complexity domains such as Shopping, CMS, and Multisite. These findings further validate the robustness of WebOperator .

Table 5: Success rate (SR %) comparison on WebArena.

| Agent | Model | Overall (#812) | Reddit (#106) | GitLab (#180) | Shopping (#187) | CMS (#182) | Map (#109) | Multisite (#48) |
|---|---|---|---|---|---|---|---|---|
| BrowserGym | gpt-4 | 15.0 | 20.2 | 19.0 | 17.2 | 14.8 | 25.5 | - |
| LM-TS | gpt-4o | 19.2 | 11.3 | 13.9 | 27.8 | 16.5 | 26.6 | 16.7 |
| Go-Browse | qwen-2.5-7b | 22.6 | 30.7 | 15.3 | 22.4 | 25.3 | 17.9 | - |
| AWM | gpt-4 | 35.5 | 50.9 | 31.8 | 30.8 | 29.1 | 43.3 | - |
| Branch-n-Browse | gpt-4o | 35.8 | 50.9 | 36.7 | 34.6 | 26.4 | 46.8 | 18.8 |
| WebPilot | gpt-4o | 37.2 | 65.1 | 39.4 | 36.9 | 24.7 | 33.9 | - |
| AgentOccam | gpt-4-turbo | 45.7 | 67.0 | 43.3 | 46.2 | 38.9 | 52.3 | 16.7 |
| AgentSymbiotic | claude-3.5 | 52.1 | 66.0 | 51.0 | 48.0 | 49.0 | **60.0** | 29.0 |
| ScribeAgent | gpt-4o | 53.0 | 73.7 | **59.7** | 45.8 | 37.9 | 56.3 | - |
| **WebOperator** | gpt-4o | **54.56** | **76.42** | 52.78 | **49.20** | **54.95** | 55.24 | **31.25** |

## C.2 GENERALIZATION TO REAL-WORLD WEBSITES.

As shown in Table 6, WebOperator achieves 63.57% accuracy on the WebVoyager subset, surpassing AgentOccam (48.84%). Improvements are most pronounced on knowledge-intensive or structurally complex websites such as ArXiv (+31.25%) and Hugging-Face (+17.65%), indicating strong robustness to deep navigation and multi-step decision-making. Additionally, WebOperator avoids catastrophic failures seen in AgentOccam (e.g., BBC News: 0.00% → 50.00%), demonstrating more reliable behavior under ambiguity. Performance saturates on straightforward transactional sites such as Amazon and Booking, where both methods achieve near-perfect accuracy. Conversely, on direct retrieval sites like Google Search, AgentOccam performs slightly better, highlighting a trade-off: WebOperator excels in decision-heavy environments but may incur overhead when the optimal action path is trivial.

Table 6: Accuracy (%) of AgentOccam and Web-Operator on WebVoyager-subset.

| Website | AgentOccam | WebOperator |
|---|---|---|
| Allrecipes (#4) | 50.00 | **75.00** |
| Amazon (#1) | **100.00** | **100.00** |
| Apple (#7) | **28.57** | **28.57** |
| ArXiv (#16) | 31.25 | **62.50** |
| BBC News (#2) | 0.00 | **50.00** |
| Booking (#2) | **100.00** | **100.00** |
| Cambridge Dict (#9) | 66.67 | **77.78** |
| Coursera (#2) | 50.00 | **100.00** |
| ESPN (#10) | 30.00 | **30.00** |
| Google Map (#9) | 22.22 | **44.44** |
| Google Search (#16) | **81.25** | 75.00 |
| Huggingface (#17) | 47.06 | **64.71** |
| Wolfram Alpha (#34) | 52.94 | **70.59** |
| Overall (#129) | 48.84 | **63.57** |

## D SEARCH ALGORITHM

**Initialization**  Start from the initial state $s_0$ with observation $o_0$. Initialize the search tree $\mathcal{T}$ with root node $s_0$[3], and the frontier $\mathcal{F}$ as a priority queue of unexecuted actions, initially empty.

**Main Loop**  At each iteration:

1. **Expansion:** If the current state $s_c$ has not been expanded, generate candidate actions $a_i$ from observation $o_c$ using the LLM agent.

2. **Scoring:** Score each action using the process reward model **without execution**. Unlike prior works (Koh et al., 2024; Zhang et al., 2025a) that rely on executing actions and scoring the resulting states—an approach often limited to simulated environments or reversible actions.

3. **Merging:** Merge equivalent actions (e.g., clicking the same element) by aggregating their scores (summing rewards) to reflect combined promise.

4. **Tree Update:** For each merged action $a_i$, add it as an outgoing edge from $s_c$ in $\mathcal{T}$ (target state pending execution).

---

[3]As we don't have access to the full state, we store only the accessible states (or observation) in each tree node

5. **Frontier Update:** Add the merged actions to $\mathcal{F}$.

6. **Action-Aware Selection:** Select the most-promising action $a^*$ from $\mathcal{F}$ using a multi-criteria approach that considers both reward score and action type (e.g., prioritizing navigation actions over data entry).

7. **Termination:** If $a^*$ is a terminating action, return the solution trajectory from $s_0$ to $s_{\text{origin}}$ in $\mathcal{T}$.

8. **Backtracking:** If $s_{\text{origin}} \neq s_c$, attempt backtracking using speculative execution. If backtracking is infeasible (due to dynamic content changes or missing UI elements), remove $a^*$ from $\mathcal{F}$ and repeat Step 6.

9. **Execution:** Execute $a^*$ in the browser, resulting in new state $s_{\text{new}}$ with observation $o_{\text{new}}$.

10. **Tree Expansion:** Add $s_{\text{new}}$ to $\mathcal{T}$ and link $a^*$ to $s_{\text{new}}$.

11. **State Transition:** Set $s_c \leftarrow s_{\text{new}}$ and continue.

The main search procedure is detailed in Algorithm 1, which uses the queue management, destructive action detection, action selection policy and backtracking functions defined in Algorithms 4, 2, 5 and 3.

## E  BACKTRACKING

Backtracking is a crucial operation in tree search for web automation. In our search algorithm, actions are selected based on a priority function that estimates their potential utility. As a result, the agent may need to execute an action that was generated from a previous state rather than the current one. In WebOperator, we define **backtracking** as the process of navigating the agent from the current state to a previous state where a promising action was originally proposed. This process generally involves two steps: (1) restoring the environment to a state corresponding to an ancestor of the target state, and (2) executing the sequence of actions required to reach the target state.

**Ideal Backtracking.**  In an ideal scenario, to backtrack from a current state $s_c$ to a target state $s_t$, the agent would:

1. Identify the **lowest common ancestor (LCA)**[4] of $s_c$ and $s_t$ in the search tree.

2. **Undo all actions** from $s_c$ to the LCA, restoring the environment to the LCA state.

3. **Replay the actions** from the LCA to $s_t$, reaching the target state.

**Challenges of Action Undo.**  Undoing actions in web environments is often non-trivial:

- **Safe actions** (affecting temporary state) can typically be reversed.
- **Destructive actions** (modifying persistent state) may not have a defined inverse.
- Many actions, such as clicks, form submissions, or dynamic content changes, are **non-deterministic**, making perfect reversal unreliable.

As a result, executing ideal backtracking directly in the main environment is not feasible.

**Naive Backtracking.**  Prior work (Koh et al., 2024) restores the root state by reseting the environment and replays the entire sequence of actions to reach the target state. While straightforward, this approach is inefficient, as it requires executing all actions from the root to the target state.

**Optimized Backtracking.**  WebOperator introduces an improved backtracking mechanism that addresses both efficiency and reliability concerns.

- **Efficiency.** To reduce replay overhead, WebOperator employs *checkpoint-based state jumping*. A state is marked as a checkpoint when its webpage observation remains unchanged under refresh and its URL differs from that of its parent. Such states are safe to revisit directly because they are refresh-stable and represent distinct navigation points, ensuring that jumping to their URL reliably reconstructs the same underlying environment without requiring prior UI interactions.

---

[4]The lowest common ancestor is the deepest node that is an ancestor of both $s_c$ and $s_t$.

During backtracking, the agent jumps to the closest checkpoint of $s_t$ via URL navigation, then replays only the minimal UI interactions (e.g., scrolling, form filling) needed to reconstruct $s_t$. This strategy avoids unnecessary intermediate steps by leveraging the determinism of URL-based navigation: navigate to the nearest checkpoint, then apply only the incremental UI operations required to recover the exact target state.

- **Reliability.** To handle non-deterministic behaviors, WebOperator employs *speculative backtracking* with *snapshot validation*. Instead of replaying actions directly in the main environment, the agent attempts backtracking in a parallel browser tab. Before executing each stored action, the agent compares the current observation in the parallel tab with the corresponding stored snapshot of the target state $s_t$. If any mismatch indicates that the state is no longer reproducible—due to randomness, dynamic content changes, or UI drift—the backtracking attempt is immediately aborted, leaving the main environment unchanged. If all actions replay successfully and all snapshots match, the reconstructed state is committed to the main environment. This speculative execution prevents unintended side effects and ensures reliable state restoration even in non-deterministic web environments.

  A key challenge arises with actions such as `tab_focus(index)`, whose behavior depends on tab indices that may differ between the actual and speculative environments. To address this, WebOperator dynamically remaps tab indices to maintain consistency. For instance, when simulating a backtrack from a state with $N$ open tabs, an action like `tab_focus(index)` in the original sequence is transformed into `tab_focus(N + index)` (using relative indexing) to align with the temporary tab configuration.

Together, these mechanisms make backtracking both efficient and reliable, allowing the agent to reach target states while minimizing unnecessary actions and handling web environment dynamics.

## F  ACCESSIBILITY TREE-BASED OBSERVATION COMPARISON.

To robustly verify UI state equivalence during backtracking, WebOperator compares accessibility tree (AX tree)[5] structures with semantic flexibility instead of requiring exact structural matches, which would fail on real-world dynamic webpages.

**Pivotal Node.** When WebOperator performs an action (e.g., clicking a button, typing into a text field, choosing a menu option), it identifies the corresponding node for that UI element in the AX tree of the current observation. This node is designated as the pivotal node. During normal forward navigation, WebOperator stores each AX tree snapshot and records the pivotal node for the upcoming action.

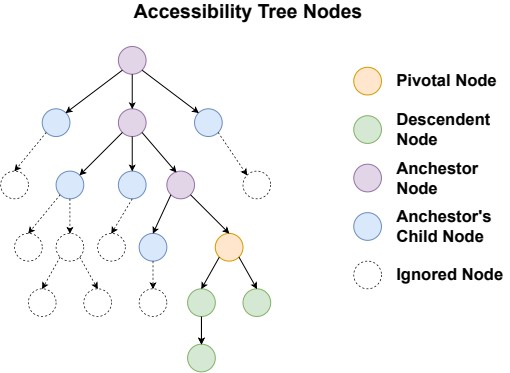

**Accessibility Tree Nodes**

- Pivotal Node
- Descendent Node
- Anchestor Node
- Anchestor's Child Node
- Ignored Node

Figure 8: Pivotal node and its context in the accessibility tree

**Backtracking Verification.** When backtracking to a previous observation, the system must ensure that the previously executed action is still valid in the current page state. To do so, it checks whether the pivotal node still exists and remains semantically equivalent. The comparison proceeds in three steps:

1. **Node Identity Check:** Using the unique node identifier recorded earlier, WebOperator checks whether a node with the same ID is present in the current AX tree. If missing, observations differ, backtracking mismatch.

2. **Node Equivalence Check:** If the node exists, WebOperator compares the semantic attributes of the node (e.g., role, label, value, enabled/disabled state). If not equivalent, mismatch.

3. **Contextual Neighborhood Check:** For additional stability, WebOperator compares the pivotal node's local structural context:

---

[5]The accessibility tree is a structured representation of a webpage's elements, exposing their roles, states, and properties to assistive technologies and programmatic agents.

- its ancestors
- its own descendants
- children of its ancestors

This localized region (illustrated in Fig. 8) ensures that the UI around the action-critical element has not significantly changed in a way that could invalidate the action. If the context matches, observations are considered equivalent.

## G ERROR ANALYSIS

We identify key factors limiting WebOperator's performance, stemming from both evaluation framework constraints and inherent autonomous web navigation challenges.

**Action Selection and Termination Challenges.**

- **Incorrect Termination:** Success hinges on the final action; incorrect reward assignment or lack of productive actions can lead to premature or wrong termination.
- **Destructive Action Side Effects:** Destructive actions can invalidate states critical to correct solutions. Safety thresholds help but cannot prevent all such dead ends.

**Task Ambiguity and Domain Knowledge.** Performance is constrained by interpreting ambiguous instructions, especially for niche domains. LLM-based rephrasing improves clarity but is limited by the model's general knowledge, leading to possible misinterpretations and failures.

**Web Interface Complexity and Feedback.** Non-intuitive UI elements and delayed or uninformative feedback (e.g., generic error pages, Figure 9) hinder diagnosis and recovery, complicating the agent's task execution.

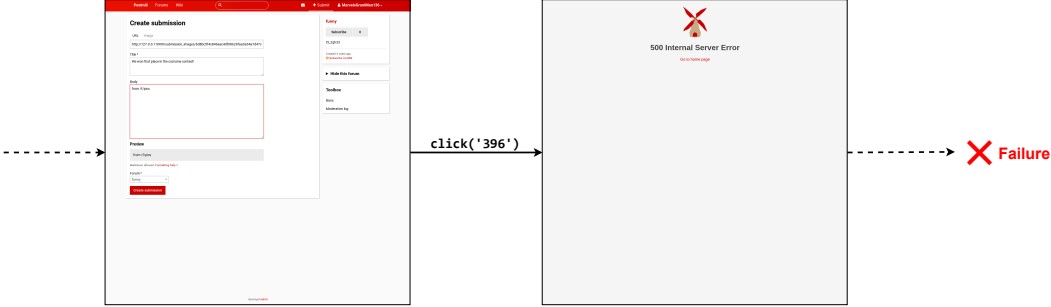

Figure 9: Filling up the URL field in reddit submission page, leads to "500 Internal Server Error". Which is really hard to understand because the error is not immediate. Agent first fills up URL field, then title, then body and finally submits the post. And then gets the error. So, it is hard to know which step causes the error. Agent tries this again and again, and eventually fails.

## H QUALITATIVE EXAMPLE

Figure 10 shows a qualitative walkthrough of our tree-search process for the task `webarena.421`.

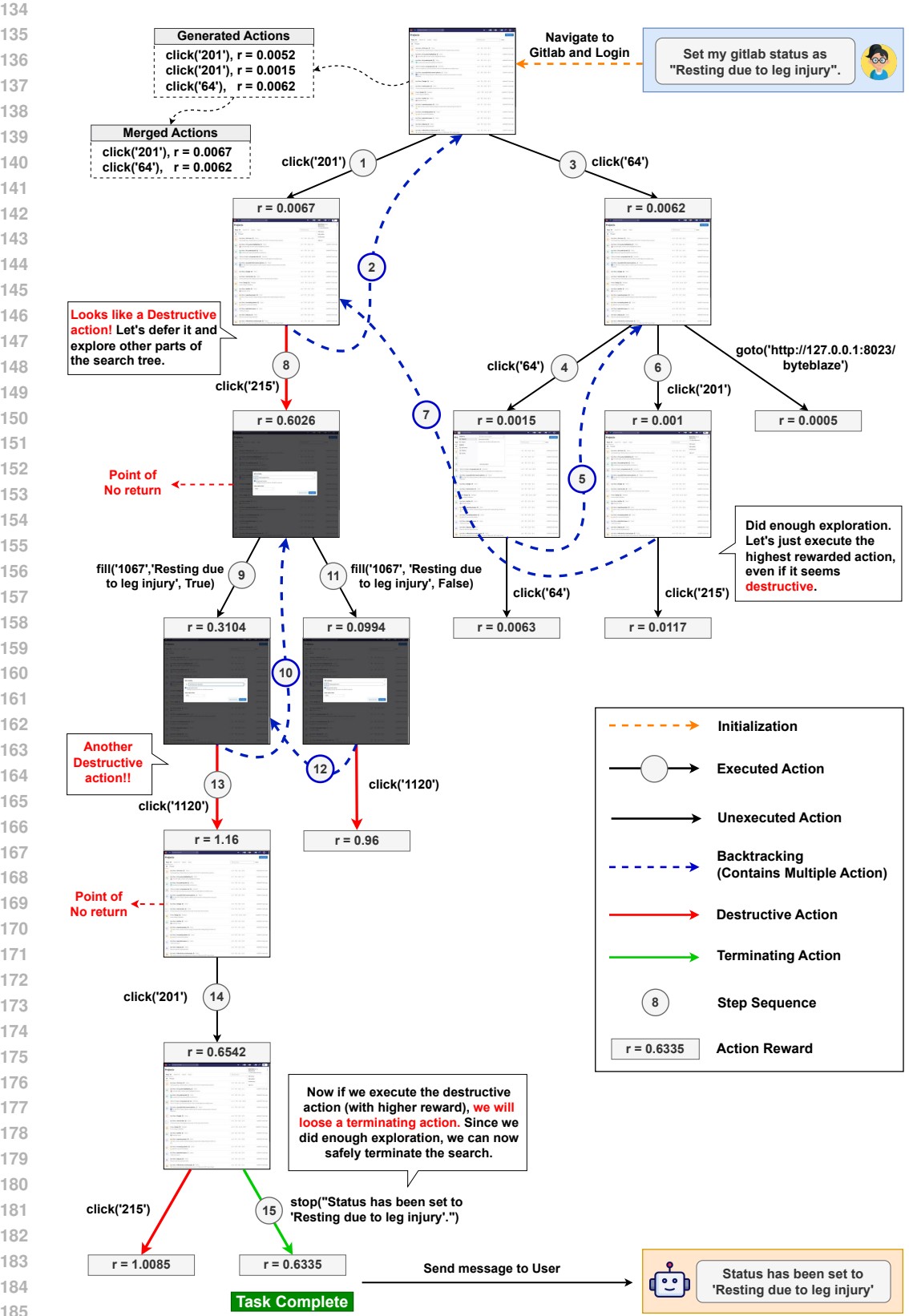

Figure 10: Step-by-step tree search corresponding to the overview in Figure 1 (Task: webarena.421).

---

**Algorithm 1** WebOperator Tree Search Algorithm

---

**Require:** Initial state $s_0$, instruction $I$, max depth $D_{\max}$, search budget $B$
**Ensure:** Task completion status

1:   $\mathcal{Q} \leftarrow \emptyset$                    ▷ Max priority queue for candidate actions
2:   current_node $\leftarrow$ create_node($s_0$)            ▷ Initialize tree with root node
3:   terminating_count $\leftarrow 0$
4:   destruction_count $\leftarrow 0$
5:
6:   **while** $\mathcal{Q}$ is not empty **do**
                                      ▷ STEP 1: Generate actions and add to queue
7:      **if** depth(current_node) $< D_{\max}$ **then**
8:         candidates $\leftarrow$ GENERATE_ACTIONS(current_node, $I$)
9:         candidates $\leftarrow$ MERGE_ACTIONS(candidates)
10:        **if** there is any terminating action in candidates **then**
11:           terminating_count $\leftarrow$ terminating_count $+ 1$
12:        **end if**
13:        $\mathcal{Q}$.push_all(candidates)
14:      **end if**
                                        ▷ STEP 2: Select best action from queue
15:      **while** $true$ **do**
16:        $(n_{selected}, a_{\text{selected}}) \leftarrow$ SELECT_ACTION($\mathcal{Q}$)
17:        **if** current_node $\neq n_{selected}$ **then**             ▷ Need to backtrack first
18:          **if** not BACKTRACK(current_node, $n_{selected}$) **then**
19:            **continue**          ▷ Backtracking failed, select another action
20:          **else**
21:            current_node $\leftarrow n_{selected}$
22:          **end if**
23:        **end if**
24:        **break**
25:      **end while**
26:      **if** $|\mathcal{Q}| > B$ **then**
27:        PRUNE_QUEUE($\mathcal{Q}, B$)                    ▷ Enforce search budget
28:      **end if**
                                        ▷ STEP 3: Execute action and adjust tree
29:      $s_{\text{new}} \leftarrow$ execute_in_env($a_{\text{selected}}$)
30:      new_node $\leftarrow$ create_node($s_{\text{new}}$)
31:      add_edge(current_node, new_node, $a_{\text{selected}}$)
32:      current_node $\leftarrow$ new_node
33:
34:      **if** is_terminating($a_{\text{selected}}$) **then**
35:        **return true**                       ▷ Task completed successfully
36:      **end if**
37:      **if** is_destructive($a_{\text{selected}}$) **then**
38:        $\mathcal{Q} \leftarrow \emptyset$                         ▷ Reset queue on destructive actions
39:        destruction_count $\leftarrow$ destruction_count $+ 1$
40:        $B \leftarrow \max(B - 1, \text{MIN\_QUEUE\_SIZE})$         ▷ Reduce search budget
41:      **end if**
42: **end while**
43: **return false**                               ▷ Task failed (queue empty)

---

**Algorithm 2** Destructive Action Detection

---

**Require:** Current action `action`, authentication status
**Ensure:** Boolean flag indicating whether `action` is destructive
 1: **if** user is not authenticated **then**
 2:     **return** False
 3: **end if**
 4: **if** `action.type` is `click` **then**
 5:     **if** selected element is not a button **then**
 6:         **return** False
 7:     **end if**
 8:     **if** button label indicates navigation or transient action (e.g., "back", "search", "refresh", "export") **then**
 9:         **return** False
10:     **end if**
11:     **if** button has popup or is disabled **then**
12:         **return** False
13:     **end if**
14:     **return** True                              ▷ Otherwise, treat as potentially destructive
15: **else if** `action.type` is `fill` and `action.args.press_enter` is True **then**
16:     **return** True
17: **end if**
18: **return** False

---

**Algorithm 3** Speculative Backtracking

---

 1: **function** BACKTRACK(current_node, target_node)
 2:     path, rollback_node ← FIND_PATH(current_node, target_node)
 3:     Open new tabs to match the rollback node and focus on the active tab of the rollback node.
 4:     **for** each action $a$ in path **do**
 5:         EXECUTE_ACTION($a$)
 6:         $o_{\text{actual}}$ ← GET_OBSERVATION()
 7:         $o_{\text{expected}}$ ← GET_STORED_OBS(node_after($a$))
 8:         **if** $not$ COMPARE_OBSERVATION($o_{\text{actual}}$, $o_{\text{expected}}$) **then**
 9:             Close new tabs and focus back on the original tab.          ▷ Discard Backtracking
10:             **return** false
11:         **end if**
12:     **end for**
13:     Close old tabs.                                     ▷ Commit Backtracking
14: **end function**
15:
16: **function** COMPARE_OBSERVATION($o_{\text{actual}}$, $o_{\text{expected}}$)
17:     pivotal_node ← GET_PIVOTAL_NODE($o_{\text{expected}}$)
18:     **if** $not$ NODE_EXISTS($o_{\text{actual}}$, pivotal_node) **then**
19:         **return** false
20:     **end if**
21:     **if** $not$ COMPARE_NEIGHBORHOOD($o_{\text{expected}}$, $o_{\text{actual}}$, pivotal_node) **then**
22:         **return** false
23:     **end if**
24:     **return** true
25: **end function**

---

---

**Algorithm 4** Core Functions for Action Management

---

1: **function** GENERATE_ACTIONS(node, instruction $I$, queue $\mathcal{Q}$)
2:     **Access global variables** terminating_count
3:     actions $\leftarrow$ GENERATE_CANDIDATE_ACTIONS($node, I$)
4:     candidates $\leftarrow \emptyset$
5:     **for** each candidate action $a$ in actions **do**
6:         $r \leftarrow f_v(I, \text{trajectory}, a)$             ▷ Process reward model prediction
7:         $c \leftarrow$ GET_PRIORITY_CLASS($a$)      ▷ 3: safe, 2: destructive, 1: terminating
8:         candidates.push($(node, a, r, c)$)
9:     **end for**
10: **end function**
11:
12: **function** GENERATE_CANDIDATE_ACTIONS(node, instruction $I$)
13:     $o \leftarrow$ get_observation(node)
14:     $H \leftarrow$ get_history($node$)
15:     $E \leftarrow$ retreive_examples($I, o$)
16:     $I_R \leftarrow$ rephrase($I$)
17:     candidates $\leftarrow \emptyset$
18:     **for** each variation in INPUT_VARIATIONS($I, H, E, I_R$) **do**
19:         **for** $i = 1$ **to** MAX_RETRY **do**
20:             $a \leftarrow$ generate_action(variation)
21:             valid $\leftarrow$ VALIDATE_ACTION(node, $a$)
22:             **if** valid **then**
23:                 candidates $\leftarrow$ candidates $\cup \{a\}$
24:                 **break**
25:             **end if**
26:         **end for**
27:     **end for**
28:     **return** candidates
29: **end function**
30:
31: **function** PRUNE_QUEUE(queue $\mathcal{Q}$, budget $B$)
32:     **if** $|\mathcal{Q}| \leq B$ **then**
33:         **return**
34:     **end if**
35:     Group actions in $\mathcal{Q}$ by priority class $c$
36:     **for** each priority class $c$ in $(1, 2)$ **do**
37:         Keep only the action with the highest reward $r$ in class $c$
38:     **end for**
39:     **while** $|\mathcal{Q}| > B$ **do**
40:         Find and remove the action with the smallest reward $r$ from $\mathcal{Q}$
41:     **end while**
42: **end function**

---

---

**Algorithm 5** Action Selection Policy

---

1: **function** SELECT_ACTION(queue $\mathcal{Q}$)
2:      **Let** $K_T$ be the threshold for discovered terminating actions
3:      **Let** $K_D$ be the threshold for executed destructive actions
4:      **Access global variables** terminating_count, destruction_count
5:      deferred_queue $\leftarrow \emptyset$
6:
7:      **if** $|\mathcal{Q}| > B$ **Or** COUNT_DESTRUCTIVE_ACTIONS($\mathcal{Q}$) $> 1$ **then**
8:          **while** $\mathcal{Q}$ is not empty **do**
9:              $(n, a, r, c) \leftarrow \mathcal{Q}.\text{pop}()$
10:             **if** $c = 1$ & terminating_count $< K_T$ **then**
11:                 deferred_queue.push($(n, a, r, c)$)
12:             **else if** $c = 2$ **then**
13:                 $\mathcal{Q}.push\_all$(deferred_queue)
14:                 **if** $\exists (a, r, c) \in \mathcal{Q} \mid c = 1$ & destruction_count $\geq K_D$ **then**
15:                     $(n^*, a^*, r^*, c^*) \leftarrow \arg\max_{(n,a,r,c) \in \mathcal{Q}, c=1} r$
16:                     $\mathcal{Q} \leftarrow \mathcal{Q} \setminus \{(n^*, a^*, r^*, c^*)\}$
17:                     **return** $(n^*, a^*)$
18:                 **end if**
19:                 **return** $(n, a)$
20:             **else**
21:                 $\mathcal{Q}.push\_all$(deferred_queue)
22:                 **return** $(n, a)$
23:             **end if**
24:         **end while**
25:     **end if**
                                          $\triangleright$ Process safe actions and terminating actions meeting threshold
26:     **while** $\mathcal{Q}$ is not empty **do**
27:         $(n, a, r, c) \leftarrow \mathcal{Q}.\text{pop}()$                 $\triangleright$ Get highest rewarded action
28:         **if** $c = 3$ **then**                 $\triangleright$ Safe action - execute immediately
29:             **return** $(n, a)$
30:         **else if** $c = 1$ & terminating_count $\geq K_T$ **then**     $\triangleright$ Terminating meeting threshold
31:             **return** $(n, a)$
32:         **else**                     $\triangleright$ Defer other actions
33:             deferred_queue.push($(n, a, r, c)$)
34:         **end if**
35:     **end while**
36:     **if** deferred_queue is empty **then**
37:         **return** None                     $\triangleright$ No action found
38:     **end if**
                        $\triangleright$ Special case for terminating actions when destruction count is high
39:     $\mathcal{Q} \leftarrow$ deferred_queue
40:     **if** $\exists (a, r, c) \in \mathcal{Q} \mid c = 1$ & destruction_count $\geq K_D$ **then**
41:         $(n^*, a^*, r^*, c^*) \leftarrow \arg\max_{(n,a,r,c) \in \mathcal{Q}, c=1} r$     $\triangleright$ Find best terminating action
42:         $\mathcal{Q} \leftarrow \mathcal{Q} \setminus \{(n^*, a^*, r^*, c^*)\}$         $\triangleright$ Remove from queue
43:         **return** $(n^*, a^*)$
44:     **end if**
                          $\triangleright$ Execute any remaining destructive action
45:     **if** $\exists (a, r, c) \in \mathcal{Q} \mid c = 2$ **then**
46:         $(n^*, a^*, r^*, c^*) \leftarrow \arg\max_{(n,a,r,c) \in \mathcal{Q}, c=2} r$     $\triangleright$ Find best destructive action
47:         $\mathcal{Q} \leftarrow \mathcal{Q} \setminus \{(n^*, a^*, r^*, c^*)\}$         $\triangleright$ Remove from queue
48:         **return** $(n^*, a^*)$
49:     **end if**
                          $\triangleright$ Execute any remaining terminating action
50:     $(n, a, r, c) \leftarrow \mathcal{Q}.\text{pop}()$             $\triangleright$ Get highest rewarded action
51:     **return** $(n, a)$
52: **end function**

---

# I  PROMPT

**Agent Prompt.**    Listing 1 shows the general prompt template for action generation. A full example with all components is available at Listing 7.

**Rephraser Prompt.**    Listing 3 shows the prompt used to rephrase a task instruction.

**WebShepherd Prompt.**    Listing 4 shows the prompt used for checklist generation and Listing 6 shows the prompt used for reward estimation.

Listing 1: Action generation prompt template

```
You are an expert web automation agent that performs precise actions on web pages to
accomplish user goals. Your task is to analyze the current page state and select the single
best next action.

>> Instructions
Review the current state of the page and all other information to find the best possible next
action to accomplish your goal. Your answer will be interpreted and executed by a program,
make sure to follow the formatting instructions.

{input_specifications}

>> Action Space

You are ONLY allowed to use the following action commands.

{action_space}

>> Generate the response in the following format:

# Observation Description
Describe the current page state and extract key information relevant to the goal. Focus on:
1. CONTENT EXTRACTION: If this page contains information needed for the final answer, extract
and record it explicitly. This step is crucial for ensuring the final answer is accurate and
complete. Don't miss any critical details.
2. RELEVANT ELEMENTS: Identify interactive elements, data, or content that helps accomplish
the objective
Format your observation to help future answer extraction by being specific about:
- Exact values, numbers, prices, names, dates found on the page
- Location of critical information (which sections, forms, tables contain target data)

# Reason
Explain your rationale clearly. If the current interface appears to fulfill your objective,
consider:
- Could there be hidden alternatives?
- Is there ambiguity in what's being shown (e.g., default sort orders)?
- Would it be safer to explore before committing?
Be cautious. It's OK to say: "This appears correct, but I want to confirm it by checking X."
Analyze previous actions. Do not get stuck in a loop by repeatedly trying the same action.

# Action
Select your action here. Strictly adheres to the given format. **Only issue one single action
**.
```

Listing 2: Instruction rephraser prompt

```
You are a rephraser module that helps a task-solving agent better understand a web-based task.

Given a task description, you should:
1. Rephrase the task in clearer language without changing its meaning.
2. Identify logical subgoals needed to accomplish the task.
3. Detect any vague or ambiguous terms.
4. Identify possible implicit assumptions that need verification.
5. List cautions - things the agent should be skeptical about when interpreting the task or
interacting with the interface.

Do not rely on any specific knowledge of websites, domains, or UI layouts. Stay general-
purpose.

>> Task Objective
{task_description}

>> Generate the response in the following format:
# REPHRASED TASK:
<Clear, literal paraphrase of the original task. No assumptions.>
```

```
# SUBGOALS:
- <Subgoal 1>
- <Subgoal 2>
- ...

# AMBIGUITIES OR ASSUMPTIONS:
- <Potentially vague or implied elements in the task>

# CAUTIONS:
- <Any points where the agent should validate its understanding rather than assume>
```

Listing 3: Example rephraser output for the task "Set my gitlab status as 'Resting due to leg injury'."

```
# REPHRASED TASK:
Change the user's status message in the GitLab application to the text "Resting due to leg
injury".
# SUBGOALS:
- Verify that the agent is currently authenticated on GitLab or can log in.
- Locate the user profile or status editing interface within GitLab.
- Enter the exact status text "Resting due to leg injury".
- Save or confirm the status change.
- Confirm that the status display reflects the new text.
# AMBIGUITIES OR ASSUMPTIONS:
- Which GitLab instance or URL is being used (e.g., gitlab.com, a self-hosted server).
- Whether the user is already logged in or needs authentication.
- The exact UI element that allows status editing (profile menu, settings page, sidebar, etc.)
.
- Whether there is a character limit, special formatting, or required emoji to display the
status.
- Whether the task refers to the personal status or a project/team status.
# CAUTIONS:
- Do not assume the status field is always visible; first verify its presence.
- Check that the status message "Resting due to leg injury" does not exceed any length limits
or contain disallowed characters.
- Ensure that the change is applied to the correct user account-verify the identity before
editing.
- After saving, confirm visually that the status update appeared; don't just rely on a
confirmation dialog.
- Be aware of any permissions or two-factor authentication that might block status changes.
```

Listing 4: Checklist generation prompt

```
You are an AI assistant tasked with generating structured checklists that highlight key
subgoals necessary to complete a task.

# Task Description
Generate a checklist which are key milestones for achieving the given instruction. First,
provide a concise
subgoal analysis in a single paragraph summarizing the required interactions. Then, based on
this, generate the checklist with brief description.

Note: If the target website requires login, assume the user is already logged in and starts
from an authenticated session. So, don't include any login steps in the checklist.

# Given Information
## User Instruction
{intent}

## Current State
### Current URL
{start_url}

### AXTREE
Note: [bid] is the unique alpha-numeric identifier at the beginning of lines for each element
in the AXTree. Always use bid to refer to elements in your actions.
{text_observation}
```

Listing 5: Example generated checklist for "Set my gitlab status as 'Resting due to leg injury'."

```
Checklist 1: Log into Your GitLab Account
- Ensure you are signed into your GitLab account to access account settings.

Checklist 2: Navigate to Account Settings
- Locate and open the account menu to find the option to manage your GitLab status.

Checklist 3: Set GitLab Status
- Access the section where you can set your GitLab status and select \"Resting due to leg
injury\" to apply your custom status.
```

### Listing 6: Judge prompt

```
You are an expert evaluator of web agent. Your task is to assess how helpful a given agent's
THOUGHT and ACTION is in making progress toward the user's goal, based on the current state of
 the webpage.

# Task Description
Evaluate how well the agent's THOUGHT and ACTION satisfy each item in the checklist using the
task instruction, trajectory (including previously completed steps), current webpage state,
the agent's latest response and checklist completion after (n-1)th step. Start by writing a
concise paragraph summarizing the agent's overall performance. Refer to the reasoning provided
 in the trajectory, and discuss whether the THOUGHT is appropriate and the ACTION moves the
task forward.
Then, assess each checklist item individually using the following labels:
- Yes: The item is fully and clearly satisfied, either in the current response or previously
completed.
- In Progress: There is meaningful partial progress toward completing the item.
- No: The item is not satisfied due to ambiguity, insufficient evidence, or lack of progress.

# Given Information
## User Instruction
{intent}
## Trajectory
{trajectory}
## AXTREE
Note: [bid] is the unique alpha-numeric identifier at the beginning of lines for each element
in the AXTree. Always use bid to refer to elements in your actions.
{text_observation}
## Checklist
{checklist}
## Last Step Checklist Completion
{progress}
## Agent's Response
THOUGHT: {thought}
ACTION: {action}
```

### Listing 7: Example Prompt with all components

```
>> Instructions
Review the current state of the page and all other information to find the best possible next
action to accomplish your goal. Your answer will be interpreted and executed by a program,
make sure to follow the formatting instructions.

>> Goal
Set my gitlab status as \"Resting due to leg injury\".

# REPHRASED TASK:
Change the user's status message in the GitLab application to the text "Resting due to leg
injury".
# SUBGOALS:
- Verify that the agent is currently authenticated on GitLab or can log in.
- Locate the user profile or status editing interface within GitLab.
- Enter the exact status text "Resting due to leg injury".
- Save or confirm the status change.
- Confirm that the status display reflects the new text.
# AMBIGUITIES OR ASSUMPTIONS:
- Which GitLab instance or URL is being used (e.g., gitlab.com, a self-hosted server).
- Whether the user is already logged in or needs authentication.
- The exact UI element that allows status editing (profile menu, settings page, sidebar, etc.)
.
- Whether there is a character limit, special formatting, or required emoji to display the
status.
- Whether the task refers to the personal status or a project/team status.
# CAUTIONS:
- Do not assume the status field is always visible; first verify its presence.
- Check that the status message "Resting due to leg injury" does not exceed any length limits
or contain disallowed characters.
- Ensure that the change is applied to the correct user account-verify the identity before
editing.
- After saving, confirm visually that the status update appeared; don't just rely on a
confirmation dialog.
- Be aware of any permissions or two-factor authentication that might block status changes.

>> Previous Actions and Observations
Observation #1: (Url: http://127.0.0.1:8023/)
The current view is the GitLab dashboard page. In the header navigation there is a link with
id `201` labeled \"Byte Blaze\" next to the avatar image. No status or profile editing
controls are visible on this dashboard view. The GitLab user status can be edited from a
profile or avatar menu, which is typically accessed by clicking the user avatar link.
Reason #1: To set a status, we must first navigate to the user's profile area where the status
 widget is available. The only accessible element that leads to the profile is the avatar link
```

```
1566
1567        `201`. No other interactive elements on the dashboard provide a direct way to edit the status
           .
1568       Action #1: click('201')

1569
1570       >> Current step: 2

1571       >> Currently open tabs
1572       Tab 0 - Projects · Dashboard · GitLab (http://127.0.0.1:8023/) -> Active

1573       >> Current page Accessibility Tree
1574       RootWebArea 'Projects · Dashboard · GitLab', focused, url='http://127.0.0.1:8023/'
1575           banner ''
                   [52] link 'Skip to content', url='http://127.0.0.1:8023/#content-body'
1576               StaticText 'GitLab'
                   [58] link 'Dashboard', url='http://127.0.0.1:8023/'\n\t\t[xxx] list ''
1577                   listitem ''
1578                       [64] button '', hasPopup='menu', expanded=False
                   list ''
1579                   listitem ''
1580                       [130] textbox 'Search GitLab'
                           StaticText '/'
1581               list ''
                   listitem ''
1582                       [141] link 'Create new...', url='http://127.0.0.1:8023/projects/new'
1583                   listitem ''
                           [153] link 'Issues', url='http://127.0.0.1:8023/dashboard/issues?
1584                       assignee_username=byteblaze'
                   listitem ''
1585                       [157] link 'Merge requests', url='http://127.0.0.1:8023/dashboard/
1586                       merge_requests?assignee_username=byteblaze'
                   listitem ''
1587                       [171] link 'To-Do List', url='http://127.0.0.1:8023/dashboard/todos'
1588                           StaticText '2'
                   listitem ''
1589                       [175] link 'Help', url='http://127.0.0.1:8023/help'
1590                           StaticText 'Help'
                   listitem ''
1591                       [201] link 'Byte Blaze', focused, expanded=True, url='http://127.0.0.1:8023/
1592                       byteblaze'
                           image 'Byte Blaze', url='https://www.gravatar.com/avatar/99
1593                           a4297c867eada2606b9b6973f081f9?s=48&d=identicon'
1594                       list ''
                           listitem ''
1595       .......
1596
1597       >> Similar Examples from Past Successes
1598       Here are 5 similar successful examples from past interactions:
           Example 1 (relevance: 8009.89):
1599           Goal: View the source code of the PyTorch-GAN project
1600           Thought: I have filled in the search box with 'PyTorch-GAN', but I need to submit this
              search to see the results. Therefore, I will press the 'Enter' key to initiate the search.
1601           Action: press('1038', 'Enter')

1602
1603       Example 2 (relevance: 7979.707):
              Goal: Go to your user profile page to manage account settings.
1604           Thought: The user's avatar image was clicked, but it seems to be opening the user's
              personal page rather than the profile page. Let's look for a more explicit link to the
1605          profile page. The menu list in the dropdown contains an option to 'Edit profile', which is
1606           likely to be the link to the user's profile page. I will click on this option to navigate
               to the profile.
1607           Action: click('213', button='left', modifiers=[])
1608       Example 3 (relevance: 7967.268):
              Goal: View your user profile
1609           Thought: The user has already clicked on their profile icon 'Byte Blaze', which is the
1610          first step to view their user profile. After clicking on the profile icon, the next step
              would be to click on the 'Edit profile' link to view the profile page.
1611           Action: click('213')
1612       Example 4 (relevance: 7967.268):
              Goal: View your user profile
1613           Thought: The user has clicked on their name 'Byte Blaze' to navigate to their profile. The
               next step is to view the user's profile. Since the name was clicked, it's expected that
1614          the user's profile page is now loaded. No further action is required to view the user's
1615          profile.
1616           Action: noop()
           Example 5 (relevance: 7965.269):
1617           Goal: View your user profile
1618           Thought: The user's profile is now visible, as evidenced by the additional profile
              information such as the profile picture and name. No further action is needed to view the
1619          user's profile. The next step is to notify the user that their profile is now visible.
```

```
    Action: stop('Your user profile is now visible. You can explore your profile information
    here.')
Use these examples as guidance, but adapt your action to the current context and goal.

>> Action Space
You are ONLY allowed to use the following action commands.

click(bid: str): To click on an element with its numerical ID on the webpage. E.g., `click('
a51')`.

select_option(bid: str, option: str): To select an option in a <select> element. You can
specify option value or label to select. E.g., `select_option('237', 'Option 1')`. In case
directly clicking an option returns error, you can try this out.

fill(bid: str, value: str, press_enter_after: bool): To type content into a field with a
specific ID. Note that, this function overwrites the existing text in that field. Optionally,
it can press Enter after typing. E.g., `fill("237", "example value", True)` or `fill("237", "
example value", False)`. In case this fill action is related to content writing (e.g., Reddit
posts, comments, tweets, blog posts, forum posts, reviews, messages, emails, bios,
descriptions), ensure that the "value" matches EXACTLY the text specified in the goal. Because
 your actions will be evaluated by exact string matcher.

scroll(direction: str): To navigate the webpage content. E.g., `scroll('up')` or `scroll('down
')`. In case, a page is too long vertically, some elements may be hidden. They can be revealed
 by scrolling. Such hidden items are usually represented using StaticText '[' and StaticText
']'

goto(url: str): Navigate to a url. E.g., `goto('http://www.example.com')`. It is recommended
not to try any random url unless you have discovered it in previous interations.

new_tab(url: str): Open a new tab and navigate to a url. It will become the active tab.
Example: `new_tab('http://www.example.com')`

tab_focus(index: int): Bring tab to front (activate tab). Example: `tab_focus(2)`

tab_close(): Close the current tab. Example: `tab_close()`

go_back(): To return to the previously viewed page. E.g., `go_back()`.

go_forward(): Navigate to the next page in history. E.g., `go_forward()`.

stop(text: str): To stop interaction. E.g., `stop("Based on the results of my search, the city
 was built in 1751.")`. If the task isn\'t a QnA, and you have completed the task, you should
call "stop" with appropriate message. But if the task is a QnA, you should ensure that the "
text" parameter is consistent with the "goal" and observations. Because your answer will be
evaluated by exact string matcher.

>> Generate the response in the following format:

# Observation Description
Describe the current page state and extract key information relevant to the goal. Focus on:
1. CONTENT EXTRACTION: If this page contains information needed for the final answer, extract
and record it explicitly. This step is crucial for ensuring the final answer is accurate and
complete. Don't miss any critical details.
2. RELEVANT ELEMENTS: Identify interactive elements, data, or content that helps accomplish
the objective
Format your observation to help future answer extraction by being specific about:
- Exact values, numbers, prices, names, dates found on the page
- Location of critical information (which sections, forms, tables contain target data)

# Reason
Explain your rationale clearly. If the current interface appears to fulfill your objective,
consider:
- Could there be hidden alternatives?
- Is there ambiguity in what's being shown (e.g., default sort orders)?
- Would it be safer to explore before committing?
Be cautious. It's OK to say: "This appears correct, but I want to confirm it by checking X."
Analyze previous actions. Do not get stuck in a loop by repeatedly trying the same action.

# Action
Select your action here. Strictly adheres to the given format. **Only issue one single action
**.
```

