# OpenReview forum: "WebOperator: Action-Aware Tree Search for Autonomous Agents in Web Environment"
_ICLR.cc/2026/Conference — Submitted to ICLR 2026_

### Official Review · Reviewer_KdXS · 2025-10-29

**Soundness:** 2
**Presentation:** 2
**Contribution:** 2
**Rating:** 4
**Confidence:** 4

**Summary:**

The paper proposes WebOperator, a holistic framework for web navigation that treats browser interaction as a controlled search process rather than simple next-action prediction. At each step, the system generates multiple candidate actions, scores them with a process reward model, filters out invalid or unsafe actions, prioritizes safe and reversible actions over potentially destructive ones, and uses simulation-backed backtracking to recover before committing irreversible changes. The authors report that this holistic framework outperforms prior web agents on WebArena and WebVoyager, even when using an open-weight backbone.

**Strengths:**

- The paper directly targets a well-known, real weakness of current web agents: irreversible, high-impact actions in partially observable web environments.
- The method includes a safety-aware control layer (deferring destructive actions, simulating rollback, backtracking only after verification) rather than relying purely on greedy next-step policies.
- The authors report higher success rates than prior systems on WebArena and WebVoyager, in some cases even against agents using proprietary LLMs.

**Weaknesses:**

- **Limited technical novelty**: The paper seems much more like a holistic engineering stack than a focused research contribution. The claimed contribution is presented as a large bundle (“holistic exploration strategy”) with many different parts (retrieval, instruction rewriting, candidate action generation, WebShepherd-style scoring, validation, safety tagging, destructive-action deferral, simulation-based backtracking, etc.), but the paper does not cleanly isolates one core idea or the concept of the proposed framework. Furthermore, each of the individual components proposed in this paper(e.g., retrieval of similar trajectories, action-space pruning, checklist-based action scoring, simulation backtracking) has already been widely explored (e.g., [1], [2], [3], [4], [5], [6]) not only in general agent domain but also in web navigation tasks. As a result, it’s hard to tell what is fundamentally new, which specific mechanism is essential, or where the performance actually comes from.

- **Heavy dependence on WebShepherd-style process reward**: The framework appears to lean critically on an off-the-shelf checklist/reward model (WebShepherd) for action scoring. However, it is unclear (i) whether other baselines would also gain large improvements if given the same reward model, and (ii) how much of WebOperator’s gains remain if that reward model is removed or weakened. In other words, is WebOperator itself responsible for the improvement, or is it mainly “strong process reward + safety heuristics”?

- **Missing cost/latency analysis despite a clearly more expensive pipeline**: Because WebOperator stitches together many stages (multiple candidate expansions per step, action validation, backtracking, prioritized search, etc.), one would expect significantly higher inference cost and latency compared to simpler baselines. However, the paper does not report any experiments or analysis related to this. Furthermore, if the proposed framework requires substantially more computation than existing baselines, it remains unclear whether the performance gain is large enough to justify such additional cost and latency, or whether the method is actually more efficient in practice. Additional experiments or clarification from the authors regarding cost and efficiency would strengthen the submission.

- **Lack of specified explanation of the baselines and fairness of the comparison**: The paper claims to outperform proprietary-model baselines on WebArena using only an open-source backbone, but some of those baselines are not using the same reasoning backbone (i.e, GPT-OSS-20B vs non-reasoning proprietary models), which makes the comparison hard to interpret. It is not clear whether baselines were re-run with equivalent prompting, retrieval, instruction rewriting, WebShepherd-scoring, or action validation — or whether only WebOperator benefits from those enhancements. Without this, it’s difficult to attribute gains to the proposed framework itself rather than stronger scaffolding around it.

- **Over-claiming robustness / generalizability**: The paper sometimes suggests robustness “in the wild,” but all evaluation is still within WebArena/WebVoyager-style environments. It’s not convincingly shown that the full framework generalizes to truly open, uncontrolled web settings. Given how heuristic and component-heavy the method is, it’s not obvious that it would seamlessly transfer. (e.g., to claim generalizability in the wild, shouldn’t WebOperator demonstrate that it performs well even in environments that are more in the wild than existing WebArena-like settings such as [7], [8], and [9]? The current benchmarks it compares against are not truly in-the-wild—they are benchmarks on which most other web navigation methods are already shown to work, and thus cannot fully support the claimed generalization.)

[1] HiAgent: Hierarchical Working Memory Management for Solving Long-Horizon Agent Tasks with Large Language Model, ACL’25

[2] AgentOccam: A Simple Yet Strong Baseline for LLM-Based Web Agents, arXiv’25

[3] Web-Shepherd: Advancing PRMs for Reinforcing Web Agents, NeurIPS’25

[4] WebRollback: Enhancing Web Agents with Explicit Rollback Mechanisms, arXiv’25

[5] Is Your LLM Secretly a World Model of the Internet? Model-Based Planning for Web Agents, arXiv’25

[6] Web Agents with World Models: Learning and Leveraging Environment Dynamics in Web Navigation, ICLR’25

[7] ASSISTANTBENCH: Can Web Agents Solve Realistic and Time-Consuming Tasks?, EMNLP’24

[8] Online Mind2web: An Illusion of Progress? Assessing the Current State of Web Agents, arXiv’25

[9] WorkArena++: Towards Compositional Planning and Reasoning-based Common Knowledge Work Tasks, NeurIPS’24

**Questions:**

1. Key contribution
- What is the conceptual novelty that the author wants to mainly claim, beyond “a holistic stack that works well”?
2. Claims about Process reward / WebShepherd in line 46
- The intro argues that off-the-shelf step-level usefulness models (like WebShepherd-style process rewards) are inherently short-sighted and imperfect, yet WebOperator still uses exactly such a model to score and prioritize actions. I’m a little confused in this sentence, since the weboperator itself adopts the webshephered. Can you further clarify this statements or provide concrete evidence that your framework actually mitigates that myopia (e.g., cases where WebShepherd gave a misleading score but the search / destructive-action deferral / backtracking logic recovered)? Or quantitative results showing meaningful gains even when that reward model is ablated?
3. Cost and scalability.
- Can you please provide per-task end-to-end time of web operator and the compared baselines, or any kind of latency/computation analysis? Is WebOperator actually practical at scale, or are we trading significantly higher inference cost for higher success rate?

---

> ### Author Response · Authors · 2025-11-24
>
> We thank the reviewer for the thoughtful feedback.
>
> > W5: Over-claiming robustness / generalizability
>
> Our evaluation on real-world websites via the WebVoyager benchmark was included precisely to demonstrate transfer beyond simulated environments. However, we focused our analysis primarily on WebArena because it contains the destructive, write-based tasks that are the central challenge our method addresses. WebVoyager, while a valuable real-world benchmark, consists largely of read-only tasks, which limits its utility for evaluating our core contributions like destruction-aware backtracking. Therefore, we position our results as demonstrating strong generalizability across established benchmarks, with WebArena providing the critical test for destructive actions and WebVoyager confirming operation on real websites.
>
> ---
>
> Other raised concerns have been addressed in the updated manuscript, as detailed in the common response above.

---

### Official Review · Reviewer_Eubc · 2025-10-31

**Soundness:** 3
**Presentation:** 3
**Contribution:** 2
**Rating:** 4
**Confidence:** 4

**Summary:**

The paper introduces WebOperator, a tree-search based web-agent framework that selects actions using an action aware, best-first search strategy. It shows that WebOperator with open source LLMs can outperform other agent approaches using closed-source LLMs on WebArena and WebVoyager.

WebOperator combines various techniques, some of which build upon prior works, which when put together attains SOTA results on WebArena benchmark for open models. It,
1. Refines the task instruction to make it less ambiguous
2. Performs observation space pruning: Extend AgentOccam [1] to filter redundant observations and prune infeasible actions.
3. Performs action consolidation: Merge redundant or semantically similar actions into a unified representation for reward computation
4. Reduces computational overhead in backtracking: Extend Tree-Search by Koh-et.al [2] and adopt a best-first search strategy. They make backtracking efficient by using URL-based jumps to the closest ancestor and replaying only the minimum required actions. They also handle destructive actions by heuristically detecting destructive actions and marking them as a point of no return in the search process.
5. Reuses experience: Use RAG for candidate action generation and validation
6. Uses the existing WebShepherd reward model to generate a list of sub-goals for an instruction. This checklist guides the reward signal generation which controls the tree search.

[1] AgentOccam: A Simple Yet Strong Baseline for LLM-Based Web Agents (https://arxiv.org/abs/2410.13825)

[2] Tree Search for Language Model Agents (https://arxiv.org/pdf/2407.01476)

**Strengths:**

1. WebOperator combines many techniques which when put together achieve state-of-the-art results on WebArena (55.68%) with an open-source backbone LM, outperforming baselines using closed-source LLMs like AgentSymbiotic, ScribeAgent etc.(leaderboard can be found here: https://docs.google.com/spreadsheets/d/1M801lEpBbKSNwP-vDBkC_pF7LdyGU1f_ufZb_NWNBZQ/edit?gid=0#gid=0)
2. On WebVoyager, WebOperator achieves a higher success rate (63.57%)  when compared to AgentOccam (48.84%).
3. The paper implements novel techniques to tackle typical challenges of tree-search algorithms for web-agents, like improving backtracking via action simulation and detecting destructive actions.


Overall, the technical contribution is not entirely novel but they combine multiple ideas well to demonstrate strong results on web navigation benchmarks.

**Weaknesses:**

1. The approach is not original in any one of the many techniques the paper uses. Tree-search for web-agents has been explored in prior works as mentioned in the paper, and it is unclear what percentage gains are brought about by WebOperator’s novelties on top of existing works like AgentOccam [1] and Tree-search [2]. A baseline which could have been used is Tree-search algorithm by Koh et. al. with AgentOccam’s observation space improvements or Tree-search algorithm with the WebShepherd reward model.
2. It would help to have a detailed ablation study of all heuristics/techniques used in WebOperator, like instruction reframing, hyperparameters used in the tree search (like branching factor, search depth etc.), handling JavaScript alerts in the axtree, and action merging. Right now it is not clear which of these heuristics brings the most improvement.
3. The experiments in Table 2 do not help ablate gains from models v/s agent designs.
4. The results WebOperator attains on WebVoyager are not close to state-of-the-art. Since AgentOccam is the only baseline used for WebVoyager, it is not proven that WebOperator’s gains generalize to benchmarks apart from WebArena.
5. No quantitative results are provided to back the claims made in Error Analysis (Section 3.3).
6. The paper attempts to solve issues with backtracking to enable tree-search, but the techniques used are still heuristics to optimize the search instead of a clear solution for backtracking based exploration. Thus, the claim that the paper addresses key limitations of existing web agents (L480-481) is incorrect.
7. Humans navigate the web by using browser-based navigation instead of relying on isolated simulations on the environment. The practicality, feasibility or challenges of using WebOperator on real websites is not addressed in the paper.


[1] AgentOccam: A Simple Yet Strong Baseline for LLM-Based Web Agents (https://arxiv.org/abs/2410.13825)

[2] Tree Search for Language Model Agents (https://arxiv.org/pdf/2407.01476)

**Questions:**

Suggestions:
1. In table 4, Cambridge Dict max result is bolded incorrectly. The bold font for ties misrepresents the ablation study column.
2. In Figure 5, what is Tool?
3. The claim that benchmark attains state-of-the-art performance on WebVoyager (eg, L484) is incorrect. (ref: https://github.com/sagekit/webvoyager)
4. Appendix Section C (L897) is empty
Questions:
1. How are past trajectories selected for the RAG examples? (L212)
2. Can the authors determine the performance of other baseline agent frameworks on WebVoyager? AgentOccam is not previous state-of-the-art on WebVoyager to warrant a fair comparison.
3. What is the cost/latency of running this agent? How does it compare to other baseline agents?
4. How are the instructions rephrased (L197-198)? Is this a multi-turn interaction or single turn?
Can we clarify what we mean by Dynamic Environment in Table 1?

---

> ### Author Response · Authors · 2025-11-24
>
> We thank the reviewer for the thoughtful feedback. The raised concerns have been addressed in the updated manuscript, as detailed in the common response above.

---

### Official Review · Reviewer_kPe4 · 2025-11-01

**Soundness:** 3
**Presentation:** 2
**Contribution:** 2
**Rating:** 2
**Confidence:** 4

**Summary:**

WebOperator is an action-aware tree-search framework for reliable web agents. It ranks actions by both reward and safety, integrates simulation-verified backtracking to avoid irreversible errors, and refines exploration through instruction reformulation, adaptive observation, and retrieval-augmented examples. These design choices enable efficient and robust task completion in partially observable web environments. Experiments on WebArena and WebVoyager show that even with open-source LLMs, WebOperator surpasses state-of-the-art proprietary agents, achieving strong generalization and practical efficiency.

**Strengths:**

S1. Engineering Maturity and Real-World Robustness
- The system demonstrates a high level of engineering completeness and stability, functioning reliably in real browser environments (e.g., simulation tabs, URL-based backtracking).
- It effectively handles realistic constraints such as partial observability, irreversible actions, and search efficiency, showcasing strong robustness and precision under real-world conditions.

S2. Competitive and Generalized Performance
- The model achieves consistently strong results across multiple realistic benchmarks, including WebArena and WebVoyager.
- It outperforms or matches powerful proprietary baselines, demonstrating generalization beyond a single dataset and adaptability across diverse web environments.

S3. High Quantitative Performance
- Quantitatively, the model shows solid empirical results across benchmarks, supporting the effectiveness of the proposed approach in web-based reasoning and navigation tasks.

S4. Action Safety and Classification
- The paper acknowledges the risk of destructive actions and classifies action types to mitigate potential harm.
- This shows awareness of safety considerations in web-agent design, contributing to practical reliability.

**Weaknesses:**

W1. Lack of Readability and Coherence
- The paper’s writing style and organization are inconsistent, making it difficult to follow the main narrative.
- The introduction fails to clearly convey the motivation, problem statement, and core contributions.
- Key components—such as Rephrase Instruction and Optimized Observation—are insufficiently described, weakening conceptual clarity.

W2. Limited Research Novelty
- The method primarily combines existing heuristics and engineering improvements rather than introducing a fundamentally new algorithmic concept.
- Theoretical depth is limited; for instance, the URL-based backtracking closely resembles prior work such as WebRollback (https://arxiv.org/abs/2504.11788).
- The paper lacks clear articulation of what distinguishes its conceptual contributions from existing frameworks.

W3. Insufficient Quantitative Analysis of Core Features
- Although backtracking and destructive-action handling are presented as key innovations, their actual impact is not empirically validated.
- No dedicated experiments, subsets, or metrics (e.g., efficiency, success rate) directly measure these components’ contributions.

W4. Missing Ablation and Comparative Studies
- The absence of ablation analysis prevents understanding of how much each module contributes to overall performance.
- Comparisons against strong baselines (e.g., GPT-4, tree-search vs. non–tree-search methods) are limited, leaving the empirical strength of claims unsubstantiated.

W5. Efficiency and Inference Cost Concerns
- Tree-search–based reasoning likely increases inference latency and computational cost, raising concerns about practical usability.
- The paper lacks measurements or discussions of efficiency (e.g., runtime, search cost, latency).
- Without fair comparisons under equal model capacity, claims of performance gains may not justify the added computational overhead.

W6. Shallow Qualitative and Case Analyses
- The paper provides little qualitative insight into model behavior—no example traces, success/failure case studies, or interpretive discussions.
- This omission limits readers’ understanding of why the system succeeds or fails under specific conditions.

**Questions:**

- Comparison with World Model–based planning: The motivation emphasizes the necessity of tree search–based exploration, yet no direct comparison is provided against World Model–based planning approaches such as WebDreamer (https://arxiv.org/abs/2411.06559). The trade-offs between these paradigms should be clearly presented.
- Necessity of backtracking: It remains unclear whether backtracking is always needed. For instance, in a movie-booking scenario where the user mistakenly selects The Lord of the Rings instead of Harry Potter, it would be more efficient to correct the action within the current page rather than returning to the initial state. Such realistic use cases should be discussed to clarify when backtracking provides real benefit.
- Table 5 issue: The performance peak when Rephrased is disabled and Retrieved examples = 0 is unexplained. This suggests unclear roles and interactions among the system’s components and warrants further clarification.
- World Model:  Even if world model has more inference time, heuristic has lots of risk in the partially observable web environment. It is well done to classify the type of an action, isn't it better to predict the following state with world model for action classification? Why doesn’t the paper use a world model for action validation or destructiveness detection, as in prior work (https://arxiv.org/abs/2410.13232)? A learned world model could offer more general and adaptive validation than heuristic rules, so clarifying this choice would strengthen the paper’s justification.

---

> ### Author Response · Authors · 2025-11-24
>
> We thank the reviewer for the thoughtful feedback.
> >Q1: Comparison with World Model–based planning: The motivation emphasizes the necessity of tree search–based exploration, yet no direct comparison is provided against World Model–based planning approaches such as WebDreamer. The trade-offs between these paradigms should be clearly presented.
>
> The reviewer is correct that a direct comparison with world model-based planning is an interesting direction. However, our core research objective is not to argue that tree-search is universally superior, but rather to demonstrate that within this paradigm-which is particularly suited to handling irreversible actions and non-determinism-carefully designed heuristics can overcome its fundamental limitations.  A thorough empirical comparison with world-model approaches, which involve different architectural assumptions and computational budgets, is a significant undertaking that we leave for future work.
>
> >Q2: Necessity of backtracking: It remains unclear whether backtracking is always needed. For instance, in a movie-booking scenario where the user mistakenly selects The Lord of the Rings instead of Harry Potter, it would be more efficient to correct the action within the current page rather than returning to the initial state. Such realistic use cases should be discussed to clarify when backtracking provides real benefit.
>
> We thank the reviewer for this insightful point. A key contribution of our method is its checkpoint-based backtracking (detailed in common response above) mechanism, which directly addresses the limitation of resetting to the initial state. Instead of full resets, our system strategically backtracks to the most recent relevant checkpoint. This provides a more efficient recovery mechanism than full state resets while maintaining the benefits of systematic exploration. We have clarified this checkpointing approach and its advantages in the revised manuscript.
>
> >Q4: World Model: Even if world model has more inference time, heuristic has lots of risk in the partially observable web environment. It is well done to classify the type of an action, isn't it better to predict the following state with world model for action classification? Why doesn’t the paper use a world model for action validation or destructiveness detection, as in prior work? A learned world model could offer more general and adaptive validation than heuristic rules, so clarifying this choice would strengthen the paper’s justification.
>
> The choice of heuristic-based reasoning was a deliberate design decision to prioritize computational efficiency and transparency for real-time web interaction. While we agree that world models represent a powerful alternative, we respectfully disagree with the characterization of our method as carrying "lots of risk." Our framework incorporates multiple safeguards—including tree-reset on destruction, and speculative backtracking with snapshot validation (detailed in common response above)—that collectively eliminates the risks typically associated with heuristic approaches in partially observable environments. The state-of-the-art results achieved by our system demonstrate that these lightweight components form a robust and highly effective architecture.
>
> ---
>
> Other raised concerns have been addressed in the updated manuscript, as detailed in the common response above.

---

### Official Review · Reviewer_vYGy · 2025-11-01

**Soundness:** 3
**Presentation:** 3
**Contribution:** 2
**Rating:** 4
**Confidence:** 3

**Summary:**

The paper extends a best-first search approach for solving web tasks by so-called action-awareness, where a type of action-safety reasoning, action merging inside the search tree and backtracking using URL-based state jumping are proposed. The authors suggest heuristics for guiding the agent towards safer actions which are not destructive for the task at hand.

The approach is evaluated on WebArena and WebVoyager via BrowserGym. Baselines tree search methods such as LM-TS and non-tree search methods such as AgentOccam.

The results show superior performance for numerous baselines on WebArena, and better performance compared to AgentOccam on the lite variant and WebVoyager.

**Strengths:**

- The methodological improvements over LM-TS are well-motivated
- The evaluation is sufficiently thorough, focussing on WebArena.
- The evaluation results are promising achieving significant improvement with an open-source model compared to competitors on larger, commercial models.

**Weaknesses:**

- Improvements are iterative compared to LM-TS.
- Also provided methods for dealing with destructive actions are heuristics to sort-of shape rewards towards safe actions, so more generalized solutions are still needed.
- Safety is a central concept in the paper, but the related works as well as empirical evaluation do not sufficiently focus on this aspect. I think this needs to be improved before acceptance.

**Questions:**

- Are there cases where the heuristics could make the agent too conservative?
- Are there additional related works focussing on LLM agent safety in other ways, e.g., post-hoc to inference or via alignment? Could we use these too and could these perform better?

---

> ### Author Response · Authors · 2025-11-24
>
> >W1: Improvements are iterative compared to LM-TS.
>
> We respectfully disagree. LM-TS assumes deterministic, fully reversible environments and performs naïve replay-based backtracking. In contrast, WebOperator introduces several structural, non-iterative advances that address fundamental limitations of LM-TS and other tree-search agents:
> 1. **Action-aware safety framework:** We introduce a principled action taxonomy (safe, temporary, destructive, invalid) with pre- and post-execution destructiveness detection. Prior tree-search agents treat all actions as reversible and do not model environment mutability.
> 2. **Checkpoint-based speculative backtracking:** Instead of URL-only restoration, WebOperator uses refresh-stable checkpoints, tab-level simulation, and snapshot validation. This enables reliable recovery under DOM drift, dynamic rendering, and history-dependent states—conditions where LM-TS backtracking fails.
> 3. **Action-quality control pipeline:** We incorporate dynamic action-space adaptation, action validation, semantic merging, and context variation. LM-TS expands raw LLM samples, leading to redundant or invalid branches; our pipeline restructures the search space itself.
> 4. **Safety-aware search policy:** We explicitly prioritize reversible actions and defer destructive ones, maintaining frontier integrity after irreversible transitions—an ability absent in LM-TS.
> 5. **Empirical evidence of non-iterative benefit:** With only 10 search budget, WebOperator improves success rate over LM-TS by +23.5% using the same backbone, demonstrating capabilities (e.g., safe destructive action handling, reliable rollback) that LM-TS cannot realize.
>
> Overall, these components change how tree-search agents represent state, handle irreversibility, and recover from nondeterminism. They are not incremental refinements but a redefinition of the operational assumptions that LM-TS relies on.
>
> >W2: Also provided methods for dealing with destructive actions are heuristics to sort-of shape rewards towards safe actions, so more generalized solutions are still needed.
>
> We acknowledge the reviewer's point regarding the need for generalized solutions for destructive actions. Our main focus was to demonstrate that a simple heuristic-based approach can yield substantial gains in complex web environments. In response to this feedback, we have introduced a more generalized post-execution heuristic (detailed in common response above) that triggers when resetting the search tree after executing destructive actions, providing a more robust safety mechanism. While this represents an important step forward, we agree that further work on generalized safety approaches remains valuable future work.
>
> >W3: Safety is a central concept in the paper, but the related works as well as empirical evaluation do not sufficiently focus on this aspect. I think this needs to be improved before acceptance.
>
> Please refer to the common response above for updated related works.
>
> >Q1: Are there cases where the heuristics could make the agent too conservative?
>
> Our heuristics are designed to defer destructive actions, not to block them indefinitely, which prevents the agent from becoming overly conservative. The system ensures progress by executing a deferred destructive action when the search frontier is exhausted or when two destructive actions are deferred consecutively, making it a viable last resort. This design balances caution with the ability to complete tasks that require destructive steps, ensuring they are taken only when necessary.
>
> >Q2: Are there additional related works focussing on LLM agent safety in other ways, e.g., post-hoc to inference or via alignment? Could we use these too and could these perform better?
>
> We have updated the Related Works section to include discussions on alternative LLM agent safety paradigms. These methods often employ post-hoc monitoring or specialized model-based detectors to identify and mitigate risky actions. Our framework's heuristic-based approach offers a distinct advantage in computational efficiency. Integrated directly into the planning loop, it provides real-time, low-latency safety reasoning, which is crucial for maintaining the responsiveness of an interactive web agent. While the model-based methods mentioned by the reviewer offer one path to generalization, their significant computational overhead makes them less suitable for our tree-search setting. We position the exploration of such methods for higher-level safety guarantees as an exciting avenue for future work.

---

### Author Response · Authors · 2025-11-22

We thank all reviewers for their constructive and insightful feedback. We have substantially revised the main text of the paper to address concerns around novelty, safety reasoning, clarity, fairness of comparison, and empirical validation. Below we summarize the major updates made in this revised version submitted alongside the rebuttal. (_A fully polished version, including an updated appendix, will be submitted by the end of the rebuttal period._)
### **1. Fair Comparison Using a Common Backbone**

To ensure that performance differences reflect algorithmic design rather than model capacity, we now use **GPT-4o** for both action generation and scoring in WebOperator.
For baselines, we rely on results reported directly in their original papers—specifically the prior tree-search methods LM-TS, WebPilot, and Branch-n-Browse—which also use GPT-4o.
As shown in **Table 2** of the updated paper, this setup ensures that all methods are evaluated under an identical backbone, enabling a fair and controlled comparison.

### **2. Component-Level Ablation Studies**
We added a dedicated ablation section (**Sec. 4.3**) that evaluates the contribution of each WebOperator component with inference cost:
 - Dynamic Action Space
 - Action Validation
 - Context Variation
 - Action Merging
 - Destructive-Action Handling
 - Context-Aware Action Selection
 - Speculative Backtracking

These experiments (**Table 3** in updated paper) quantify the contribution of each module and demonstrate that WebOperator’s gains arise from its design rather than external factors.

### **3. Clearer Positioning of Tree-Search–Specific Contributions**

We restructured the paper to focus on the tree-search innovations absent in prior literature, separating them from general agent engineering.

### **4. Checkpoint-Based Speculative Backtracking:**

Earlier state restoration relied on URL-based navigation, but identical URLs do not guarantee identical states in real web environments. Page observations often depend on preceding actions and states—e.g., transient UI changes, form submissions, cookies, or hidden query parameters—and can be affected by DOM drift, dynamic rendering, asynchronous API calls, and session-dependent content. Consequently, URL-only restoration is unreliable.
To address this, we introduce _checkpoint-based backtracking_, absent in existing tree-search agents. As detailed in **Section 3.2**, a state is considered a _checkpoint_ only if:
  - Its observation remains consistent after a page refresh, showing robustness to nondeterminism.
  - Its grandparent state has a different URL, marking a meaningful navigation boundary.\
During backtracking, WebOperator jumps only to these validated checkpoints, ensuring reliable state recovery and avoiding the brittleness of URL-only replay.

We further introduce _speculative backtracking_ to improve efficiency compared to our previous version of simulation-verified backtracking. Instead of simulating backtracking in isolation, the agent:
- Opens a new tab in the same browser and attempts backtracking with snapshot validation.
- Closes the new tab on failure, leaving the main environment intact.
- On success, replaces old tabs with the new ones.
This reduces redundant computation and ensures safe, real-time backtracking in dynamic web environments.

### **5. More Rigorous Destructive-Action Detection:**
In addition to pre-execution safety heuristics, we added a _post-execution_ detection mechanism based on HTTP semantics. As detailed in **Section 3.3**, WebOperator monitors whether an action triggers persistent state changes by inspecting HTTP methods—specifically POST, PUT, and DELETE requests—which reliably indicate database-modifying operations. This allows robust handling of database-modifying actions—an essential step for reliable rollback and safe exploration.

### **6. Expanded Safety Reasoning and Related Work:**
We strengthened the related-works section with coverage of:
- Web-agent safety
- Rollback and reversibility
- Safety-critical and risk-aware exploration
- Tree-search reliability

The updated paper clarifies why safety reasoning is needed even when using strong reward models, and how WebOperator complements rather than duplicates them.


---

### **Summary**

Across these updates, the revised submission directly addresses the key reviewer concerns regarding:
- Fairness of evaluation
- Novelty within tree-search WebAgents
- Safety mechanisms
- Backtracking reliability
- Ablation and empirical evidence
- Clarity and technical presentation

We believe the revised paper is substantially strengthened and now presents a clearer, more rigorous, and more compelling contribution to the field.

---

### Author Response · Authors · 2025-11-26
**Authors-Reviewer Discussion**

Dear Reviewers,

Thank you once again for your valuable comments on our submission. As the discussion phase is approaching its end, we would like to confirm whether we have sufficiently addressed all your concerns. If there are any remaining questions requiring further clarification, please let us know.

We sincerely look forward to your feedback!

---

### Comment · Area_Chair_tmmB · 2025-11-28
**Reminder: Engage with Authors During Rebuttal**

Dear reviewers, please engage with the authors during the rebuttal if you haven’t yet. The deadline is approaching; add clarifications and follow-ups in the submission thread to ensure a fair, informed decision. Thank you

---

### Meta-Review · Area_Chair_rxfL · 2026-01-02

**Summary:**

The paper presents a system for LLM-based web tasks that combines a number of techniques (enumerated by reviewer Eubc), largely centered on improving an agentive tree search procedure. The system improves substantially on prior web agents on the WebArena and WebVoyager benchmarks.

Strengths:
The results are strong, outperforming strong proprietary models even with an open-source model.

The approaches to tackle tree-search algorithm challenges, such as detecting destructive actions, improving backtracking, and simulating rollback, are all well-motivated.

**Reviewer Concerns:**

Weaknesses:
A major concern of the reviewers was that while the paper is a holistic engineering stack, most of the individual components have been explored in prior work. The authors did add a component-level ablation study during the response, which is helpful to identify how much each of these components contributes to the overall performance. But, the contribution is still diffuse and less focused than it could be, so it's difficult to have a clear reusable takeaway from the paper (other than the existence of the high-performing system).

The paper would benefit substantially from a rewrite that focuses more narrowly on, for example, the tree search improvements and compares to tree search baselines, e.g. the ones suggested by reviewer Eubc. The edits made during author response, with the ablations and the new method for detecting destructive actions, are a great step towards this, but in my opinion the paper would really benefit from another round of review given how substantial the changes are and the reservations of the reviewers.

**Reviewer Scores:**

I predict that vYGy might have raised their score to 6, and KPe4 might have raised their score to 4, so that the final scores would be 6 / 4 / 4 / 4.

---

### Decision · Program_Chairs · 2026-01-26

Reject